

# Toward a method for downscaling sea ice pressure

Jean-François Lemieux[1], Bruno Tremblay[2], and Mathieu Plante[2]

[1]Recherche en Prévision Numérique Environnementale/Environnement et Changement Climatique Canada, 2121 route Transcanadienne, Dorval, Qc, Canada.
[2]Department of Atmospheric and Oceanic Sciences, McGill University, Montréal, Qc, Canada.

**Correspondence:** Jean-François Lemieux (jean-francois.lemieux@canada.ca)

**Abstract.** Sea ice pressure poses great risk for navigation; it can lead to ship besetting and damages. Contemporary large-scale sea ice forecasting systems can predict the evolution of sea ice pressure. There is, however, a mismatch between the spatial resolution of these systems (a few km) and the typical dimensions of ships (a few tens of m) navigating in ice-covered regions. In this paper, we investigate the downscaling of sea ice pressure from the km-scale to scales relevant for ships. Results show

that sub-grid scale pressure values can be significantly larger than the large-scale pressure (up to $\sim 4\times$ larger in our numerical experiments). High pressure at the sub-grid scale is associated with the presence of defects (e.g. a lead). Numerical experiments show that a ship creates its own high stress concentration by forming a lead in its wake while navigating. These results also highlight the difficulty of forecasting the small-scale distribution of pressure and especially the largest values. Indeed, this distribution strongly depends on variables that are not well constrained: the rheology parameters and the small-scale structure

of sea ice thickness (more importantly the length of the lead behind the ship).

## 1 Introduction

With the growing shipping activities in the Arctic and surrounding seas, there is a need for user relevant sea ice forecasts and products at multiple time and spatial scales. An important forecast field for navigation is the internal sea ice pressure (simply referred to as pressure for the rest of this paper). In compact ice conditions, high pressure events can complicate navigation

activities and even pose great risk for ship besetting.

With the use of constitutive equations (or rheology), sea ice models are able to predict the evolution of the pressure field. However, even for high resolution operational forecasting systems with spatial resolutions of a few km (e.g. Dupont et al. (2015); Hebert et al. (2015)), there is a clear mismatch in the spatial scales considered. Indeed, the forecast pressure from the

model, which represents the average pressure for a grid cell of a few km wide, is not necessarily relevant for a much smaller ship; there are larger pressure values than the average pressure provided by the sea ice forecasting system (Kubat et al., 2010; Leisti et al., 2011; Kubat et al., 2012). Figure 1 shows an example of a pressure forecast from a large-scale forecasting system. The Canadian Arctic Prediction System (CAPS) is a fully-coupled atmosphere-sea ice-ocean system developed and maintained by the Canadian Centre for Meteorological and Environmental Prediction. Its domain covers the Arctic Ocean, the North At-

lantic and the North Pacific. The spatial resolution of the atmospheric model is $\sim$3 km while the spatial resolution in this region

for the sea ice and ocean models is ∼4.5 km. Looking at a specific region, that is north of Svalbard (panel b), it can be observed that the surface winds push the ice toward the coast and create large pressure.

Kubat et al. (2010) conducted idealized numerical simulations of a ship transiting through a loose sea ice cover. They showed
that the pressure on the hull of the ship can be two orders of magnitude larger than the large-scale pressure. Through a parameter sensitivity study, they also demonstrated that ship velocity has the most pronounced impact on the total ice force applied on the ship.

Some researchers have also done case studies of compressive or besetting events using large-scale sea ice forecasting sys-
tems (e.g. Kubat et al. (2012); Leisti et al. (2011); Kubat et al. (2013)). These besetting events are all associated with heavy ice conditions. The investigations of Kubat et al. show the importance of the coast on pressure conditions; the sea ice pressure often increases toward the coast.

Mussells et al. (2017) used ship logs and satellite imagery to relate besetting events and density of sea ice ridges. They
indeed found that the ship was often beset in areas and times of the year with high ridge densities. Probabilistic models for ship performance in sea ice and likelihood of besetting events have also been developed (e.g. Montewka et al. (2015); Turnbull et al. (2019)). Turnbull et al. (2019) argue that the primary cause of the besetting events they studied were the relatively large ice floes encountered by the vessel.

There is also a vast literature on the performance of ships navigating in ice infested waters and on the estimation of ice resistance, that is the longitudinal forces applied on the ship by the ice. These calculations are important for ship design and for operational considerations. Lindqvist (1989) introduced a simple empirical formulation to calculate ice resistance based on ship's characteristics and ice conditions. Numerical models of ships navigating in level ice that consider processes such as crushing and bending failure have also been proposed (e.g., Su et al. (2010); Lubbad and Loset (2011); Jeong et al. (2017)).
There are also some numerical studies of ice loads on ships by representing the sea ice as discrete elements (i.e., the floes, Metrikin and Lset (2013); Daley et al. (2014)).

In this paper, we use a continuum based viscous-plastic sea ice model to investigate the downscaling of sea ice pressure from the km scale to scales relevant for navigation activities (tens of m). In a first set of simulations, we study how the small-scale
pressure depends on the stresses applied at the boundaries, on the ice conditions and on the rheology parameters. The second part of the results is dedicated to shipping applications; we investigate the small-scale pressure field in the vicinity of an idealized ship beset in heavy ice conditions and under compressive stresses.

This paper is structured as follow. In section 2, the sea ice momentum equation and the viscous-plastic rheology are de-
scribed. The sea ice model used for the numerical experiments is presented in section 3. The approach for prescribing sea



ice stresses at the boundaries is presented in section 4. A coarse-graining procedure to define sea ice conditions is described in section 5. A digitized ship is used for some numerical experiments. The implementation of this digitized ship is presented in section 6. The validation of our experimental setup is done in section 7. The main results are given in section 8. Finally, concluding remarks are provided in section 9.


## 2 Sea ice momentum equation and rheology

Large-scale sea ice forecasting system solves the sea ice momentum given by

$$\rho h \frac{D\mathbf{u}}{Dt} = -\rho h f \hat{\mathbf{z}} \times \mathbf{u} + \tau_a - \tau_w + \nabla \cdot \sigma - \rho h g \nabla H_d, \tag{1}$$

where $\rho$ is the density of the ice, $h$ is the ice volume per unit area (or the mean thickness and just referred to as thickness in this paper), $\frac{D}{Dt}$ is the total derivative, $f$ the Coriolis parameter, $\mathbf{u} = u\hat{\mathbf{x}} + v\hat{\mathbf{y}}$ the horizontal sea ice velocity vector, $\hat{\mathbf{x}}$, $\hat{\mathbf{y}}$ and $\hat{\mathbf{z}}$ are unit vectors aligned with the x, y and z axis of our Cartesian coordinates, $\tau_a$ is the wind stress, $\tau_w$ the water stress, $\sigma$ the internal ice stress tensor with components $\sigma_{ij}$ acting in the j$^{th}$ direction on a plane perpendicular to the i$^{th}$ direction, $g$ the gravitational acceleration and $H_d$ the sea surface height. This two-dimensional formulation, which is obtained by integrating along the vertical, is justified when the ratio between the horizontal and vertical scales of the problem is large (i.e., a ratio of at least 1:10, Coon et al. (1974)).

The sea ice pressure is by definition the average of the normal stresses, that is

$$p = -(\sigma_{11} + \sigma_{22})/2, \tag{2}$$

with a negative sign because, by convention, stresses in compression are negative. The sea ice pressure is the first stress invariant (i.e., it does not vary with the choice of the coordinate system). The second stress invariant ($q$), that is the maximum shear stress at a point, is defined by

$$q = \sqrt{\sigma_{12}^2 + \left[\frac{(\sigma_{11} - \sigma_{22})}{2}\right]^2}. \tag{3}$$

Because the sea ice stresses are written as a function of the sea ice velocity, one also obtains the sea ice pressure $p$ and the maximum shear stress $q$ when solving the momentum equation for $\mathbf{u}$. Hence, by solving the momentum equation for the large-scale sea ice model, the pressure at every grid point is obtained (we refer to this pressure field as the large-scale pressure).





Here, we consider a small area of sea ice (the size of a grid cell) to be under compressive stresses. The idea is to apply the large-scale pressure at the boundaries of this small area and to simulate the sub-grid scale sea ice pressure (referred to as the small-scale pressure). We assume here that the ice is not moving nor deforming e.g. it is being held against a coast). To further

simplify the problem, the wind stress, the water stress and the sea surface tilt term are neglected. This means that we wish to find the solution of $\nabla \cdot \sigma = 0$ inside this small domain. The stresses are modeled according to the viscous-plastic rheology with an elliptical yield curve (Hibler, 1979). With this rheology, the relation between the stresses and the strain rates can be written as

$$\sigma_{ij} = 2\eta\dot{\epsilon}_{ij} + [\zeta - \eta]\dot{\epsilon}_{kk}\delta_{ij} - P\delta_{ij}/2, \quad i,j = 1,2, \tag{4}$$

where $\delta_{ij}$ is the Kronecker delta, $\dot{\epsilon}_{ij}$ are the strain rates defined by $\dot{\epsilon}_{11} = \frac{\partial u}{\partial x}$, $\dot{\epsilon}_{22} = \frac{\partial v}{\partial y}$ and $\dot{\epsilon}_{12} = \frac{1}{2}(\frac{\partial u}{\partial y} + \frac{\partial v}{\partial x})$, $\dot{\epsilon}_{kk} = \dot{\epsilon}_{11} + \dot{\epsilon}_{22}$, $\zeta$ is the bulk viscosity, $\eta$ is the shear viscosity and $P$ is a term which is a function of the ice strength.

The bulk and shear viscosities are respectively

$$\zeta = \frac{P_p}{2\triangle}, \tag{5}$$

$$\eta = \zeta e^{-2}, \tag{6}$$

where $P_p$ is the ice strength, $\triangle = \left[(\dot{\epsilon}_{11}^2 + \dot{\epsilon}_{22}^2)(1 + e^{-2}) + 4e^{-2}\dot{\epsilon}_{12}^2 + 2\dot{\epsilon}_{11}\dot{\epsilon}_{22}(1 - e^{-2})\right]^{\frac{1}{2}}$, and $e$ is the aspect ratio of the ellipse, i.e. the ratio of the long and short axes of the elliptical yield curve.

Following Hibler (1979), the ice strength $P_p$ is parameterized as

$$P_p = P^* h \exp[-C(1 - A)], \tag{7}$$

where $P^*$ is the ice strength parameter, $A$ is the sea ice concentration and $C$ is the ice concentration parameter, an empirical constant set to 20 (Hibler, 1979) such that the ice strength decreases quickly with the ice concentration.

When $\triangle$ tends toward zero, equations (5) and (6) become singular. To avoid this problem, $\zeta$ is capped using an hyperbolic

tangent (Lemieux and Tremblay, 2009)

$$\zeta = \zeta_{max} \tanh(\frac{P_p}{2\triangle\zeta_{max}}). \tag{8}$$





As in equation (6), $\eta = \zeta e^{-2}$. The coefficient $\zeta_{max}$ is set to the value proposed by Hibler (1979): $2.5 \times 10^8 P_p$ (this is equivalent to limiting $\triangle$ to a minimum value of $2 \times 10^{-9} \mathrm{s}^{-1}$).

We use a replacement closure similar to the one presented in Kreyscher et al. (2000). The $P$ term in equation (4) is given by

$$P = 2\zeta\triangle. \tag{9}$$

## 3   Experimental setup

The McGill sea ice model is used for the numerical experiments. We use revision 333 with some modifications, described below, for specifying stresses at the boundaries.

Considering a domain of a few km by a few km wide (representing a grid cell of a large-scale sea ice forecasting system), the idea is to use the model at very high resolution for studying the distribution of pressure inside that domain. To do so, the model was modified so that internal stresses can be specified at the boundaries (instead of the usual Dirichlet condition (i.e. $\mathbf{u} = 0$) at land-ocean boundaries and the Neumann condition at open boundaries with gradients of velocity equal to zero). These stresses at the boundaries represent the integrated effect of the wind and ocean-ice stresses (like one would get from a

large-scale model). The next section gives more details about the implementation of the stress boundary conditions.

For the experiments, the domain is a square of dimensions 5.12 km by 5.12 km. It is subdivided in small squared grid cells of dimensions $\Delta x$ by $\Delta x$ with $\Delta x$ taking one of the following values depending on the experiment: 10 m, 20 m, 40 m, 80 m, 160 m, 320 m, 640 m or 1280 m. The size of the domain was chosen because it is close to the average size of CAPS sea ice

grid cells and because 5120 m divided by the $\Delta x$ listed above gives an integer number ($n$) of small grid cells. For simplicity, we refer to this domain as our 5×5 km domain.

The spatial discretization of $\nabla \cdot \sigma = 0$ leads to a system of nonlinear equations that is solved using a Jacobian free Newton Krylov (JFNK) solver with the most recent version described in Lemieux et al. (2014). The nonlinear convergence criterion is

reached when the Euclidean norm of the residual has been reduced by a factor of $10^6$. Thermodynamic processes and advection of $h$ and $A$ are neglected for all the numerical experiments described in this paper.

## 4   Boundary conditions

The boundary conditions are imposed the same way on the four sides of the small domain. Hence, to shorten the paper, only the

treatment on the west side of the domain is explained here. The McGill model uses an Arakawa C-grid; the center of the cell is the point for tracers (e.g. $h$ and $A$) while the velocity components are positioned on the left side (for $u$) and lower side (for $v$).





To avoid confusion with the indices $i$ and $j$ for the stresses $\sigma_{ij}$ and the strain rates $\dot{\epsilon}_{ij}$, the indices $l$ and $m$ are respectively used to identify the grid cells along the $x$ and $y$ axes. The cell at the southwesternmost location of the domain has indices $l =1$ and $m =1$. Figure 2 shows one of the grid cell on the first column of the domain (on the west side). The left side of the grid cell is

on the west boundary of the domain. The sides of the domain are referred to as west ($W$), east ($E$), south ($S$) and north ($N$).

On the west side of the domain, a normal stress ($\sigma^W$) and a shear stress ($\tau^W$) are applied. The momentum balance for the $u$ component is comprised of the terms $\partial\sigma_{11}/\partial x$ and $\partial\sigma_{12}/\partial y$. Inside the domain, these terms are approximated by second-order centered differences. At the boundaries, however, a one-sided first-order approximation is employed for $\partial\sigma_{11}/\partial x$. Hence,

$\partial\sigma_{11}/\partial x$ at the $u$ location $u(l,m) = u_{(lm)}$ with $l =1$ is approximated as

$$\frac{\partial\sigma_{11}}{\partial x} \sim \frac{\sigma_{11(1m)} - \sigma^W_{(m)}}{\Delta x/2}, \tag{10}$$

where $\sigma_{11(1m)} = [\zeta_{(1m)} + \eta_{(1m)}][u_{(2m)} - u_{(1m)}]\Delta x^{-1} + [\zeta_{(1m)} - \eta_{(1m)}][v_{(1m+1)} - v_{(1m)}]\Delta y^{-1} - P_{(1m)}/2$ is evaluated at the $t$ point.

On the other hand, the term $\partial\sigma_{12}/\partial y$ only depends on the boundary conditions, that is

$$\frac{\partial\sigma_{12}}{\partial y} \sim \frac{\tau^W_{(m+1)} - \tau^W_{(m)}}{\Delta y}, \tag{11}$$

This means that even though $u_{(1m)}$ is located at the boundary, it is also solved by the nonlinear solver.

For the $v$ component $v_{(1m)}$ (which is at a distance of $\Delta x/2$ from the boundary), there is no special treatment for $\partial\sigma_{22}/\partial y$.

However, the second-order treatment of the term $\partial\sigma_{12}/\partial x$ follows

$$\frac{\partial\sigma_{12}}{\partial x} \sim \frac{\sigma_{12(2m)} - \tau^W_{(m)}}{\Delta x}, \tag{12}$$

where $\sigma_{12(2m)} = \eta_{(2m)}[u_{(2m)} - u_{(2m-1)}]\Delta y^{-1} + \eta_{(2m)}[v_{(2m)} - v_{(1m)}]\Delta x^{-1}$ is evaluated at the $n$ point.

In our simulations, $\sigma^W_{(m)} = \sigma^W$ and $\tau^W_{(m)} = \tau^W$, i.e., they do not vary with $m$ along the boundary (same idea for the other

sides of the domain). Furthermore, for numerical stability (see appendix A), the normal stress on the east side ($\sigma^E$) has to be equal to $\sigma^W$. Similarly, $\sigma^S = \sigma^N$ and $\tau^W = \tau^E = \tau^S = \tau^N$.





## 5   Coarse-graining procedure

For some of the results, a coarse-graining procedure is used to obtain low resolutions ice conditions from $h$ and $A$ fields defined

at high resolution. This procedure is explained in Fig. 3.

## 6   Idealized ship

We have also coded in the model an idealized representation of a ship beset in heavy ice conditions. This allows us to investigate the distribution of small-scale pressure around this idealized ship. Two masks are defined: one mask that represents the

inside ($i$) of the ship at the tracer points ($M_i$) and one mask $M_c$ that defines the contour ($c$) of the ship (defined at the $n$ point as shown in Fig. 2). The mask $M_i$ is zero everywhere except at tracer points that define the shape of the ship (i.e., $M_i =1$). The mask $M_c$ is constructed from $M_i$; $M_c$ is equal to 1 if the sum of the four neighboring $M_i$ is either 1, 2 or 3.

The idea is to embed the idealized ship in the sea ice model by specifying special mechanical characteristics at the $M_i =1$

points and on the contour of the ship (i.e., where $M_c =1$). First, the strong resistance of the ship in compression is represented by setting the ice strength to $\beta P^*$ where $M_i =1$ (this is equivalent to assuming that the ship is sea ice with a thickness of $\beta$ m). For most of the experiments, $\beta$ is set to 10.0. This implies that the interior (structure) of the ship also has strong resistance to shear as the ellipse aspect ratio is also set to 2 (i.e., $e_i = e = 2$ where $M_i =1$). The maximum shear strength is in this case $0.25\beta P^*$.


The mechanical characteristics of the contour are associated with mechanical interactions between the sea ice and the ship. The model requires the calculation of the $\eta$ at the $n$ points. For the calculation of the $\eta$ on the contour of the ship, $P_p$ is assumed to be equal to the ice strength of the level ice (i.e., $P_p = h_l P^*$ which is the mechanical strength of the weakest of the two materials). Moreover, it is assumed that the ice can slide relatively easily on the side of the ship. To represent this, the shear

strength on the ship contour (i.e., where $M_c=1$) is set to a smaller value than the one for sea ice. This is done by specifying a larger ellipse ratio on the contour ($e_c$) than the value specified for sea ice ($e$) and inside the ship ($e_i$). Hence, $e_c = \kappa e$ with $\kappa > 1$ and $e =2$ for the experiments with the ship. For most of these experiments, $\kappa$ is set to 5.0.

## 7   Model validation

The McGill model has, over the years, been extensively tested (e.g. Lemieux et al. (2014); Bouchat and Tremblay (2017); Williams and Tremblay (2018)). A few simple experiments were conducted in order to validate the implementation of the new stress boundary conditions.




Compared to realistic pan-Arctic simulations, the simplicity of the problem allows one to obtain analytical solutions for

specific cases. In a first validating experiment, the thickness ($h$) and concentration ($A$) fields are respectively set to 1 m and 1 everywhere on the domain. By specifying $\sigma^W = \sigma^E = \sigma^S = \sigma^N$ = -10 kNm$^{-1}$ (i.e., $p$ =10 kNm$^{-1}$) and $\tau^W = \tau^E = \tau^S = \tau^N$ = 0 kNm$^{-1}$, at the boundaries, the shear stress should be zero everywhere inside the domain while the pressure field should be constant and equal to 10 kNm$^{-1}$. This is indeed what is obtained from the numerical experiment (not shown). With $p$ =10 kNm$^{-1}$, a 1 m ice cover is able to resist this compressive stress, that is the ice should be in the viscous regime. Using the

definition of the stresses from equation (4), we obtain $p = P/2 - \zeta \dot{\epsilon}_I$, where $\dot{\epsilon}_I = \dot{\epsilon}_{11} + \dot{\epsilon}_{22}$ is the divergence. To simplify the problem here, the replacement pressure is not used (i.e., $P = P_p$) and the simple capping approach of Hibler (1979) for $\zeta$ is employed instead of the capping given by equation 8. With these assumptions, the analytical solution is $\dot{\epsilon}_I = 5.18 \times 10^{-10}$ s$^{-1}$, which is exactly what is obtained with the model (not shown).

We also verify that we obtain the same results when a lead is present within the physical domain for different spatial resolution ($\Delta x$). For example, Fig. 4 shows the pressure field for a 1 km long, 40 m wide lead resolved with a $\Delta x$ of 10 m (a) and for the same lead resolved with $\Delta x$ =20 m (b). The maximum pressure at 10 m resolution is 31.42 kNm$^{-1}$ while the maximum pressure at 20 m is 29.09 kNm$^{-1}$. The probability density functions (PDF, Fig. 4c) also demonstrate that the simulated fields are very similar at 10 and 20 m resolutions.


The effect of the same lead but oriented differently in the domain was also tested. The PDF of the pressure field is exactly the same whether the lead is oriented horizontally (west-east) or vertically (south-north, not shown). The spatial distribution of pressure is qualitatively the same when orienting the lead diagonally. The PDF of pressure for this diagonal lead is similar to the PDF of the vertical and horizontal ones although we find that the maximum pressure is usually a bit smaller (not shown).

This is likely a consequence of the spatial discretization of a finite width lead on a cartesian grid.

In a last set of experiments for the validation, we also checked that the presence of relatively nearby boundaries do not affect our conclusions. In the first experiment, a horizontal 1.5 km long and 20 m wide lead was positioned in the center of the 5.12 km by 5.12 km domain. In a second experiment, the same lead was positioned in a domain twice this size, that is the boundaries

are much further from the lead in the second experiment. The pressure fields around the lead are very similar (not shown) in the two experiments with maximum pressures in the domain equal respectively to 22.77 kNm$^{-1}$ and 23.02 kNm$^{-1}$ (a difference of $\sim$ 1%). To avoid these boundary effects, we will tend to position the important features in the center of the domain for the numerical experiments. For a numerical experiment to be valid, we require that the simulated pressure in the first grid cells around the domain has to be within 10% of the pressure value specified at the boundaries.






## 8  Results

To limit the number of parameters than can be varied in the numerical experiments, the normal stresses at the boundaries are always equal to -10 kNm$^{-1}$ while the shear stresses are set to zero. In other words, $\sigma^W = \sigma^E = \sigma^S = \sigma^N = -10$ kNm$^{-1}$ and $\tau^W = \tau^E = \tau^S = \tau^N = 0$ kNm$^{-1}$.


In a first set of experiments, we conduct idealized experiments to investigate the impact of sea ice features (leads, ridges, etc.) on the small-scale pressure field and especially on the maximum pressure. These experiments will give us insights and guide us for the second series of experiments with the idealized ship. Fig. 5a shows the pressure field for a uniform sea ice cover with $h_l = 2$ m except the presence of a long 1 km lead. We have introduced $h_l$ with the subscript '$l'$ referring to as 'level$'$

ice. Large pressure are observed at the tips of the lead. In a second experiment, the same thickness sea ice conditions are used except that a smaller lead, a refrozen lead (with $h = 0.5$ m) and a thick sea ice ridge (with $h = 5$ m) are also positioned in the 5×5 km domain. The pressure field for this latter experiment is shown in Fig. 5b. Fig. 5c compares the PDFs of pressure for these two experiments. Looking at the PDFs and comparing Fig. 5a and Fig. 5b, one can notice that the other features are not associated with such high pressure values and that the maximum pressure is associated with the long 1 km lead. To further

support this conclusion, note that the maximum pressure in the 5×5 km domain is 38.19 kNm$^{-1}$ in the first experiment while it is 38.35 kNm$^{-1}$ in the second one. In other words, the other features do not change our analysis; what really matters is the longest lead as it is in the vicinity of the longest lead that the largest stress concentration is found.

Our results above suggest that only the longest lead needs to be considered for estimating the largest small-scale pressure.

For a given $h_l$ and stresses applied at the boundaries, there is more and more stress concentration when increasing the length of a lead. This is shown in Fig. 6 for three values of the parameter $P^*$. For short leads, the ice around the lead is able to sustain the stresses (the ice is rigid, that is in the viscous regime). This is why the three curves are very similar in Fig. 6a and Fig. 6b for short leads. However, for longer leads, there is more and more stress concentration. Some points of the ice, close to the tips of the lead, fail (i.e., the state of stress reaches the yield curve).


As the whole yield curve scales with the value of $P^*$, a larger $P^*$ leads to larger maximum pressure and shear values. When increasing $P^*$, the maximum shear stress approaches asymptotically the shear strength (dashed lines in panel a, $e^{-1}h_l P^*/2$). This asymptotic behavior is less obvious for the pressure (Fig. 6b) as it is still far from the compressive strength ($h_l P^*$). A similar behavior is observed when varying the ellipse aspect ratio (which modifies the shear strength). A smaller value of

$e$ leads to larger pressure values and larger shear stresses values (with a similar asymptotic behavior) for long leads (not shown).

While the average pressure in the domain is the same (10 kNm$^{-1}$) for all the values of $P^*$, the maximum pressure is enhanced as $P^*$ increases (as shown in Fig. 6b). Comparing the pressure fields with $P^*$=27.5 kNm$^{-2}$ and $P^*$=20 kNm$^{-2}$ (see Fig. 7) for the same lead shows that the pressure fields around the lead are different over hundreds of meters. Moreover, the





largest difference in the pressure fields are found at the tips of the lead; the pressure is much larger with $P^*$=27.5 kNm$^{-2}$ than with $P^*$=20 kNm$^{-2}$ in the vicinity of the tips.

We also investigate the evolution of the small-scale pressure field as a function of resolution. The $h$ and $A$ fields are defined at 10 m resolution. These fields $h$ and $A$ are respectively set to 2 m and 1.0 everywhere except for a 1 km long, 10 m wide lead

in the middle of the domain with $h = 0$ m and $A = 0$. The model is run at resolutions of 10, 20, 40,...1480 m. For these lower resolutions, the $h$ and $A$ fields are obtained through the coarse-graining procedure (see Fig. 3 for details). All the values of $p$ and $q$ in the 5×5 km domain are plotted as a function of $\Delta x$ in Fig. 8a and Fig. 8b. The distribution of these small-scale stresses are non-symmetric (they are limited by 0 on one side) and are skewed toward large values. These results constitute another validation of the numerical framework as the distribution reduces to a single point equal to the large-scale values prescribed at

the boundaries as $\Delta x$ tends toward the horizontal dimension of the domain.

In a second set of experiments, we investigate the small-scale pressure field in the vicinity of a ship in heavy sea ice conditions and under compressive stresses. Importantly, we estimate the maximum pressure applied on the ship in different idealized experiments. The small-scale pressure field around a ship 90 m long and 30 m wide is investigated. We assume that the ship was

navigating in level ice 2 m thick and that it is now beset. First, it is assumed that a lead created by the ship is still open behind it over a distance of 600 m while further away the lead has been closed due to resulting sea ice convergence. The pressure field for this experiment is shown in Fig. 9a and with more details in Fig. 9b. Similar to our previous results without a ship, larger pressures are found at the tips of the lead. In fact, there are very large pressure on both sides of the ship, especially at the back of the ship. Numerical simulations of ships navigating in sea ice show larger pressure at the front of the ship (e.g. Kubat et al.

(2010); Sayed et al. (2017)). However, our results show the opposite for a ship that is beset. These results also suggest that by navigating in these compact ice conditions, the ship has generated these high pressure conditions by creating a lead in its wake.

A crucial aspect to consider here is the length of the lead behind the ship. Assuming the leads closes at a shorter distance from the ship should imply smaller pressure values. This is indeed the case as it is shown in Fig. 10a. However, Fig. 10a (blue

curve) shows that even a small opening behind the ship leads to pressure values notably larger than the value prescribed at the boundaries. With only one grid cell (10 m) opened behind the ship, the pressure is ∼2.5 times the value prescribed at the boundaries. We also consider the case of a lead partially consolidated. In fact, we assume that the concentration ($A$) is 0 just behind the ship and that it increases linearly to 1.0 for a certain lead length. The (mean) thickness $h$ of the ice is set equal to $Ah_l$. This appears to have a very small effect on our results compared to the case with $A$=0 everywhere in the lead (blue curve

in Fig. 10a). This is due to the fact that the ice strength (see equation (7)) decreases rapidly as $A$ diminishes. However, if we consider that the ice in the lead is consolidating through thermodynamical growth (i.e., we set $h$ to a small value in the lead behind the ship) we find that this notably reduces the stress applied on the ship. This can be seen with the orange and magenta curves in Fig. 10a which respectively correspond to thicknesses of 0.1 m and 0.2 m for the refrozen lead.





The results above were obtained by assuming certain mechanical characteristics for the ship (compressive and shear strengths). It is physically realistic to consider large compressive and shear strengths for the structure of the ship. However, it is unclear what shear strength should be used for the contour of the ship (related to ice-ship interactions). Given level ice of thickness $h_l$ around the ship, we can reasonably assume that the shear strength for the contour of the ship has to be smaller than the shear strength of this level ice (i.e., $P^*h_l/2e$).


     Fig. 10b shows the results of a sensitivity study of the mechanical parameters for the ship. The maximum pressure applied on the ship is more sensitive to $\beta$ (associated with the compressive strength of the ship) than it is to $\kappa$ (associated with the shear strength of ship-ice interactions). See section 6 for details about $\kappa$ and $\beta$.

**9    Conclusions**

We have investigated how sea ice pressure could be downscaled at scales relevant for navigation. The distribution of pressure at small-scales is associated with non-uniform sea ice conditions. The PDF of the small-scale pressure is non-symmetric (it is limited by 0 on one side) and is skewed toward large values. Our results indicate that what really determines the largest values of pressure is associated with defects, that is long leads. Because a lead itself is not able to sustain any stress (unless it has
refrozen), the load is taken by the ice around the lead with especially large values of the stresses in the vicinity of the tips. A sensitivity study indicates that the small-scale distribution and maximum pressure are notably affected by the choice of the shear strength ($e$) and compressive strength ($P^*$) for the elliptical yield curve. This suggests that a different yield curve and different mechanical strength properties would also lead to significantly different results.

Idealized experiments with a digitized ship beset in heavy sea ice conditions show that stress concentration also occurs in the vicinity of the ship. In fact, our simulations show that the largest pressure applied on the ship is found on both sides at the back of the ship. These results are different than the ones of Kubat et al. (2010) and Sayed et al. (2017) because our idealized ship is beset while they considered a digitized ship progressing in looser ice conditions.

We also argue that the ship itself is responsible for the strong concentration of stress on its side; the lead it created by navigating in sea ice causes these large values of the stresses. Moreover, it is found that even a short lead causes pressure values notably larger than the pressure applied at the domain boundaries. The stresses on the ship should decrease as the ice in the lead consolidates (either by thermodynamical growth or closing of the lead). These conclusions highlight the difficulty of providing sub-grid scale pressure forecasts for navigation applications as the important parameters (i.e. the length of the lead and the
thickness of the refrozen ice) are not well constrained.





A significant advantage of our numerical framework is that stresses can be specified at the boundaries. However, it is also important to note its limitations. First, it can only calculate the pressure field for a ship beset in heavy sea ice conditions; it cannot simulate the sea ice stresses applied on a ship navigating in ice infested waters (as in Kubat et al. (2010)). Also, in reality, 335 sea ice convergence can cause ridging which can locally increase the yield strength of the ice. This strain hardening process was not considered in our numerical experiments; the maximum possible pressure in the domain is equal to $P^*h_l$. Finally, a possible limitation of our numerical framework is that the ice is modeled as a continuum material rather than a collection of discrete particles. It would be very interesting to still apply stresses at the boundaries but to model the interactions between the sea ice and the idealized ship with a model based on discrete floes (e.g., Daley et al. (2014); Metrikin and Løset (2013)).

*Code availability.* Revision 333 of the McGill sea ice model was modified so that stresses can be prescribed at the boundaries. This code is available on Zenodo at https://doi.org/10.5281/zenodo.3803452

## Appendix A: Stability analysis

A few observations were made concerning the numerical stability of our new numerical framework with stresses applied at the boundaries. In this appendix, we discuss and provide explanations for these limitations.

1) We have noticed that for a simulation to be numerically stable, $\sigma^W$ should be equal to $\sigma^E$, $\sigma^S$ should be equal to $\sigma^N$ and that all the shear stresses at the boundaries should have the same value (i.e., $\tau^W = \tau^E = \tau^S = \tau^N$). This can be easily understood by considering the ice in the domain as a single piece of ice. Assuming there is no shear stress, the sum of the forces applied on the ice along the $x$ axis are

$$\sum F_x = \sigma^W \Delta x - \sigma^E \Delta x. \tag{A1}$$

For stability, $\sum F_x$ should be zero so that the ice does not accelerate indefinitely. This means that $\sigma^W$ should be equal to $\sigma^E$. The same conclusion applies for $\sigma^S$ and $\sigma^N$. Finally, a similar argument can be made for the shear stresses in terms of conservation of angular momentum. Interestingly, these conditions are the same ones found for the Cauchy tensor for the stresses at a point.

2) Dukowicz (1997) mentions, that for numerical stability, the internal stresses should be zero at open boundaries while our simulations show that it is possible to obtain stable solutions with non-zero stresses prescribed at the boundaries. To understand this, we revisit the stability analysis described in Dukowicz (1997). As Dukowicz (1997), we consider a simplified





1D momentum equation. However, we also take into account the replacement pressure. With these considerations, our 1D
momentum equation is given by

$$\rho h \frac{\partial u}{\partial t} = \frac{\partial \sigma}{\partial x}, \tag{A2}$$

For stability, the rheology term should dissipate kinetic energy (KE). To investigate this, we multiply equation A2 by $u$ and integrate it over the whole domain ($x = 0$, i.e. the west side and $x = L$, i.e. the east side of our domain).

$$\int_0^L u\rho h \frac{\partial u}{\partial t} dx = \int_0^L u \frac{\partial \sigma}{\partial x} dx, \tag{A3}$$

As advection and thermodynamics are not considered, the thickness field is constant in time and we can write

$$\int_0^L \frac{\partial}{\partial t}\left(\frac{\rho h u^2}{2}\right) dx = \int_0^L u \frac{\partial \sigma}{\partial x} dx, \tag{A4}$$

In 1D, $\sigma = \alpha^2 \zeta \dot{\epsilon}_I - \zeta \Delta$ with $\zeta = \frac{P_p}{2\Delta^*}$, $\Delta^* = \min(\Delta, \Delta_{min})$, $\dot{\epsilon}_I = \frac{\partial u}{\partial x}$ and $\Delta = \alpha|\dot{\epsilon}_I|$ with $\alpha = \sqrt{1 + e^{-2}}$.

The term on the right can be integrated by parts, that is

$$\int_0^L u \frac{\partial \sigma}{\partial x} dx = [u\sigma]_0^L - \int_0^L \frac{\partial u}{\partial x} \sigma dx, \tag{A5}$$

$$\frac{\partial}{\partial t} \int_0^L \left(\frac{\rho h u^2}{2}\right) dx = u_L \sigma_L - u_0 \sigma_0 - \int_0^L \left(\alpha^2 \zeta \dot{\epsilon}_I^2 - \dot{\epsilon}_I \zeta \Delta\right) dx, \tag{A6}$$

where the time derivative has been moved outside the integral because the region of integration is fixed (Dukowicz, 1997).
Note that $u_L = u|_{x=L}$ (same idea for the other terms). The term in the integral on the left is the total KE. From our results
above we know that $\sigma_L$ has to be equal to $\sigma_0$. By symmetry, we can also assume that $u_L = -u_0$. Hence, with the definition of
the viscous coefficient, we can then write equation A6 as

$$\frac{\partial}{\partial t} KE = -2u_0 \sigma_0 - \int_0^L \frac{\alpha P_p}{2\Delta^*} \left(\alpha \dot{\epsilon}_I^2 - \dot{\epsilon}_I |\dot{\epsilon}_I|\right) dx. \tag{A7}$$

For the second term on the right, $\left(\alpha \dot{\epsilon}_I^2 - \dot{\epsilon}_I |\dot{\epsilon}_I|\right) = \dot{\epsilon}_I^2(\alpha - 1)$ if $\dot{\epsilon}_I$ is positive (divergence), while it is equal to $\dot{\epsilon}_I^2(\alpha + 1)$
if $\dot{\epsilon}_I$ is negative (convergence). As $\alpha \geq 1$, this means that the integral is always positive and the term therefore always dissipates KE because of the minus sign in front of it. As opposed to the derivation of Dukowicz (1997), the replacement pressure





is also considered here. Nevertheless, consistent with his results, we find that the second term on the right always dissipates KE.

The stability therefore depends on the boundary term $-2u_0\sigma_0$. The worst condition happens when there is strong convergence at the boundaries. In this case, $\sigma_0 = -|\sigma_0| < 0$ and $u_0 > 0$ such that $2u_0|\sigma_0|$ is a source of KE. For a large convergence, we assume that the ice is in the plastic regime. To be able to evaluate the integral on the right in equation A7, we also look at a 

simple case with $P_p$ that is constant over the whole domain. With these assumptions we find:

$$\frac{\partial}{\partial t}KE = 2u_0|\sigma_0| - \frac{\alpha P_p}{2}\int_0^L \frac{\dot{\epsilon}_I^2}{|\dot{\epsilon}_I|}dx + \frac{P_p}{2}\int_0^L \dot{\epsilon}_I dx. \tag{A8}$$

With $\dot{\epsilon}_I^2/|\dot{\epsilon}_I| = |\dot{\epsilon}_I| = -\dot{\epsilon}_I$ because $\dot{\epsilon}_I < 0$ we can then write

$$\frac{\partial}{\partial t}KE = 2u_0|\sigma_0| + \frac{\alpha P_p}{2}\int_0^L \dot{\epsilon}_I dx + \frac{P_p}{2}\int_0^L \dot{\epsilon}_I dx. \tag{A9}$$

With $\int_0^L \dot{\epsilon}_I dx = \int_0^L \frac{\partial u}{\partial x}dx = u_L - u_0 = -2u_0$ we obtain

$$\frac{\partial}{\partial t}KE = 2u_0|\sigma_0| - (\alpha + 1)P_p u_0. \tag{A10}$$

This means that $|\sigma_0|$ should be smaller that the compressive strength $(\alpha + 1)P_p/2$ for the solution to be stable (i.e., the rheology term dissipates KE). A similar analysis can be conducted if we assume a tensile stress at the boundaries. In this case, we find that the stress $|\sigma_0|$ at the boundaries should be smaller than the tensile strength $(\alpha - 1)P_p/2$.

To ensure numerical stability, Dukowicz (1997) mentions that the stresses should be zero at the open boundaries. This is a stricter condition that the one we find here. We have indeed demonstrated that the solution is stable as long as the stresses prescribed at the boundaries are between the compressive and tensile strengths of the ice. Numerical experiments (in 2D) confirm this finding. For example, when prescribing normal stresses of -10 kNm$^{-1}$ on a uniform sea ice cover, the solution is stable if $h_l > 10$ kNm$^{-1}/P^*$ (not shown).


Notice that, to base our stability analysis on the KE energy, the term $\rho h \partial u/\partial t$ had to be included. This is different than the problem that is solved in our numerical experiments (i.e., $\nabla \cdot \sigma = 0$). We have, however, verified that the same numerical solutions can be obtained by finding the steady-state solution of $\rho h \frac{\partial u}{\partial t} = \nabla \cdot \sigma$.





*Author contributions.* JFL and BT developed the downscaling method and the modified boundary conditions. JFL modified the model code and conducted the numerical simulations. JFL, BT and MP analyzed and discussed the results. JFL wrote the manuscript with contributions from BT and MP.

*Competing interests.* The authors declare no competing interest.

*Acknowledgements.* We thank Philippe Blain for his comments and for carefully reading the manuscript. B. Tremblay would like to ac-
knowledge the support of an NSERC-Discovery grant.





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



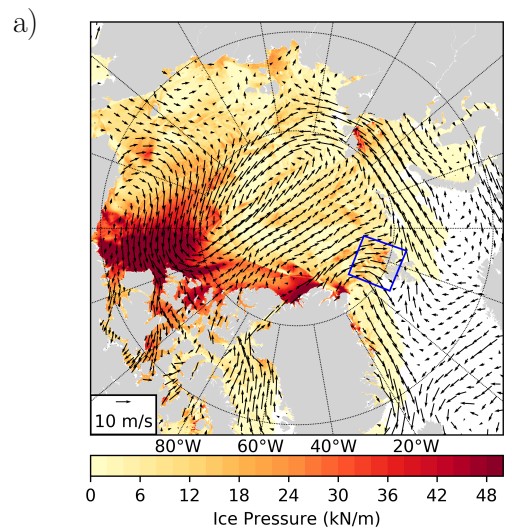

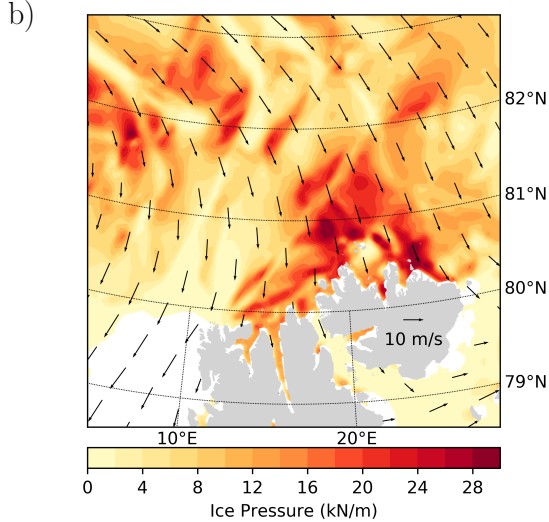

**Figure 1.** 24 h forecast of the sea ice pressure (kNm$^{-1}$) and of the surface winds (ms$^{-1}$) from the Canadian Arctic Prediction System (CAPS). The forecast was initiated at 00 UTC on 29 April 2020. Almost all of the domain is shown in panel a) while panel b) is a subset of the domain located in the region of Svalbard (the sub-region is defined by the blue rectangle in panel a). Note that the color scale is not the same for the two panels.


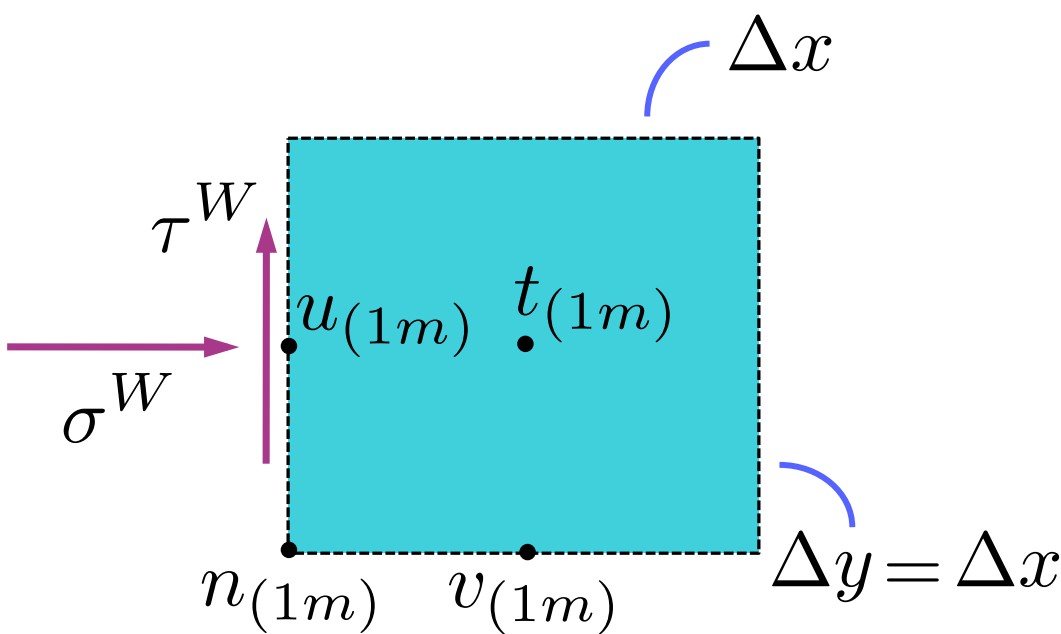

**Figure 2.** One grid cell on the western boundary of the domain with indices $l=1$ and $m$. This figure shows the location of the velocity components on the C-grid of the McGill model. The variables $h$ and $A$ are positioned at the tracer point $t$. Some variables (e.g. $\sigma_{12}$) are also calculated at the node ($n$) point. The stresses ($\sigma^W$ and $\tau^W$) applied at the western boundary are shown with purple arrows.

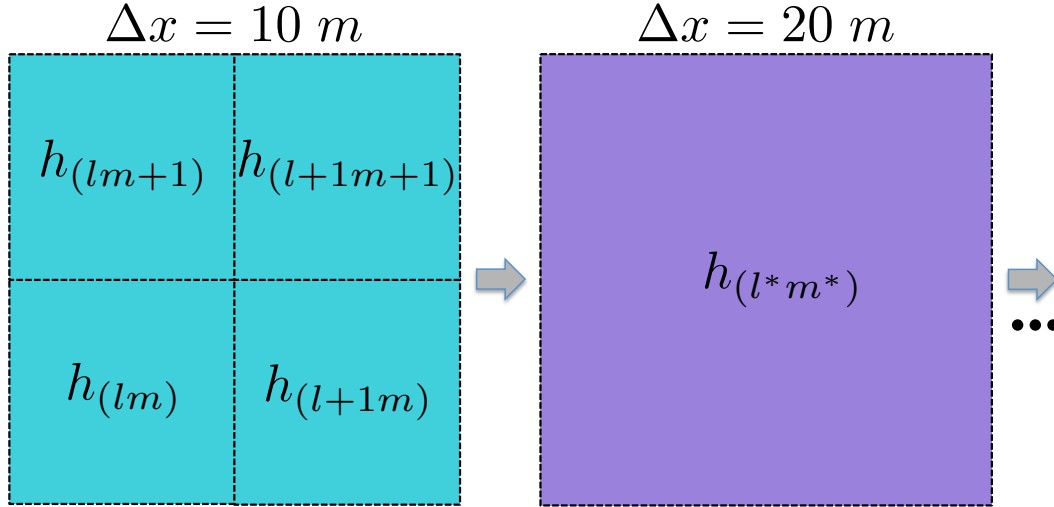

**Figure 3.** Schematic of the coarse-graining procedure. The thickness field is defined at 10 m resolution (blue cells on the left). The thickness field at 20 m resolution is obtained by averaging the $h$ values of the four 10 m cells contained in the 20 m one (purple cell). This procedure is repeated for the other lower resolutions. The same method is applied for the concentration $A$. The indices $l, m$ are for the 10 m grid while the indices $l^*, m^*$ are for the 20 m grid.

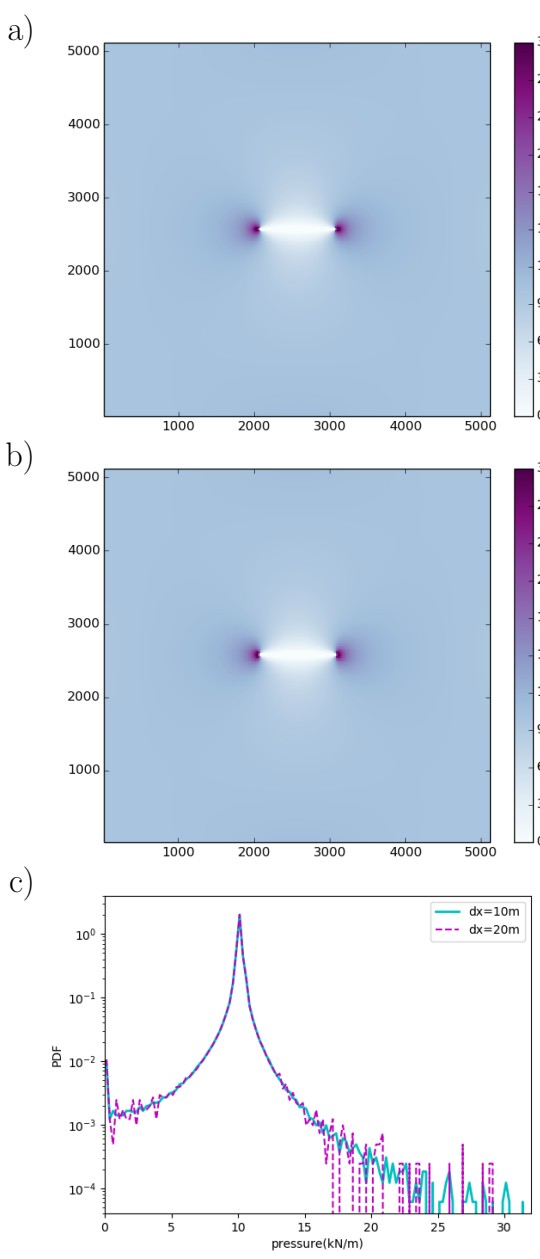

**Figure 4.** Pressure field for $\Delta x = 10$ m (a) and $\Delta x = 20$ m (b). The thickness field is 2 m everywhere except a 1 km long, 40 m wide horizontal lead in the middle of the domain. The normal stresses at the boundaries are -10 kNm$^{-1}$. The last panel (c) shows PDFs of the pressure in the 5×5 km domain for $\Delta x = 10$ m (cyan) and $\Delta x = 20$ m (magenta). Bins of 0.25 kNm$^{-1}$ were used to build the PDFs.

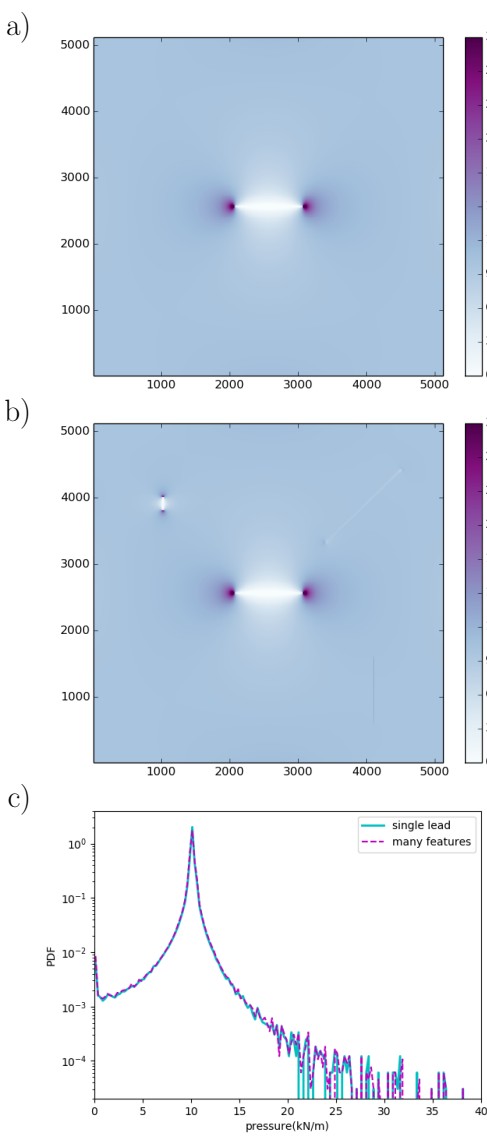

**Figure 5.** Pressure (kNm$^{-1}$) field for a thickness field of 2 m everywhere except a 1 km long, 10 m wide horizontal lead in the middle of the domain (a, referred to as 'single lead'). Pressure (kNm$^{-1}$) field for a thickness field of 2 m everywhere except a 1 km long, 10 m wide horizontal lead in the middle of the domain, a diagonal refrozen lead (h=0.5m), a smaller lead in the northwestern part of the domain and a 1 km ridge (max h = 5 m in center, 2.5 m on each side) in the southeastern part of the domain (b, referred to as 'many features'). For both experiments $\Delta x = 10$ m and the normal stresses at the boundaries are -10 kNm$^{-1}$. PDFs of the pressure for the 'single lead' experiment (cyan) and the 'many features' experiment (dashed magenta)




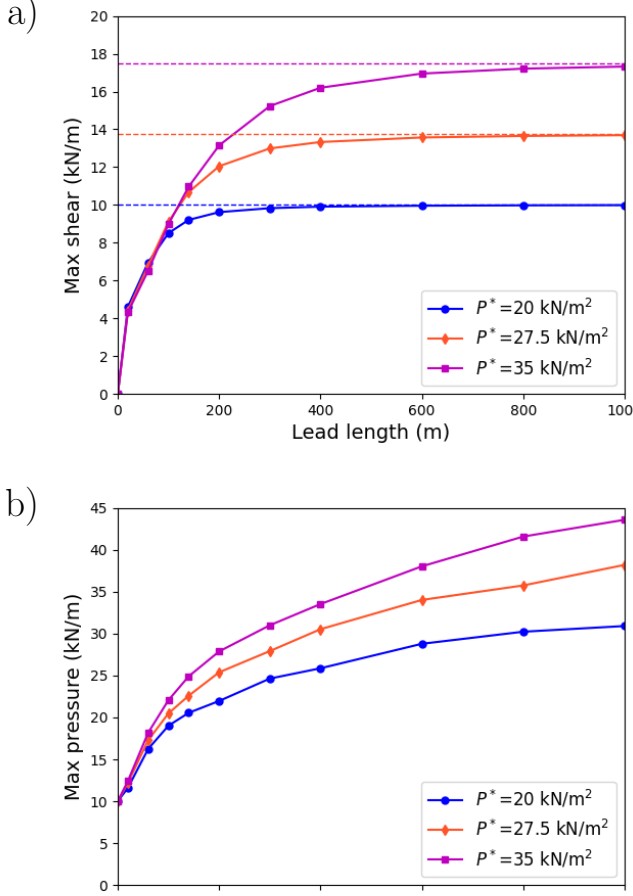

**Figure 6.** Maximum value of the shear stress invariant (a, $kNm^{-1}$) and of the pressure (b, $kNm^{-1}$) in the domain as a function of lead length for different values of the parameter $P^*$ ($P^*$=20 $kNm^{-2}$: blue, $P^*$=27.5 $kNm^{-2}$: orange, $P^*$=35 $kNm^{-2}$: magenta). The thickness field is 2 m everywhere except the 10 m wide horizontal lead in the middle of the domain. The normal stresses at the boundaries are -10 $kNm^{-1}$.



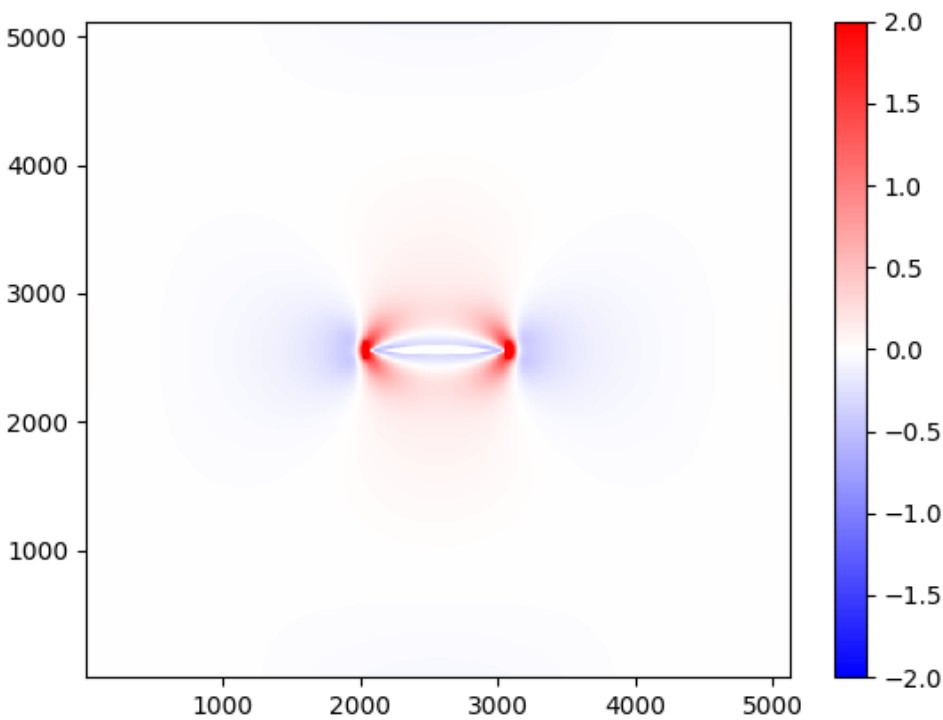

**Figure 7.** Pressure field with $P^*$=27.5 kNm$^{-2}$ minus the pressure field with $P^*$=20.0 kNm$^{-2}$ (in kNm$^{-1}$). For both experiments, the thickness field is 2 m everywhere except a 1 km long, 10 m wide horizontal lead in the middle of the domain. The normal stresses at the boundaries are -10 kNm$^{-1}$.

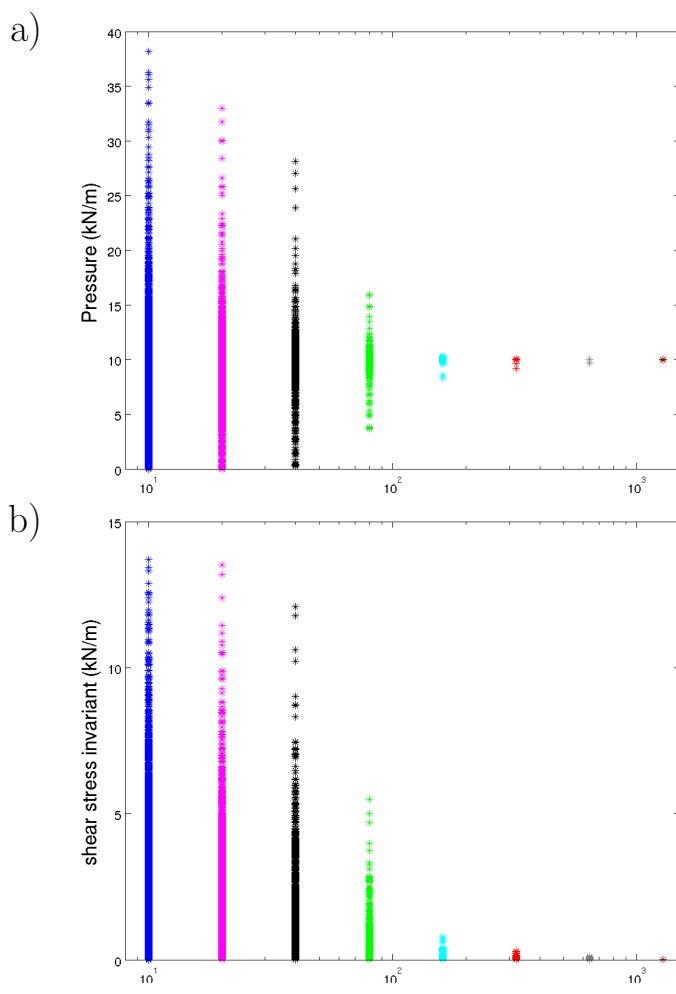

**Figure 8.** All the values of pressure (a) and of the shear stress invariant (b) in the 5×5 km domain as a function of resolution. The thickness field is 2 m everywhere except a 1 km long, 10 m wide horizontal lead in the middle of the domain. The initial thickness and concentration fields at the other resolutions are obtained through a coarse-graining procedure. The normal stresses at the boundaries are -10 kNm$^{-1}$.

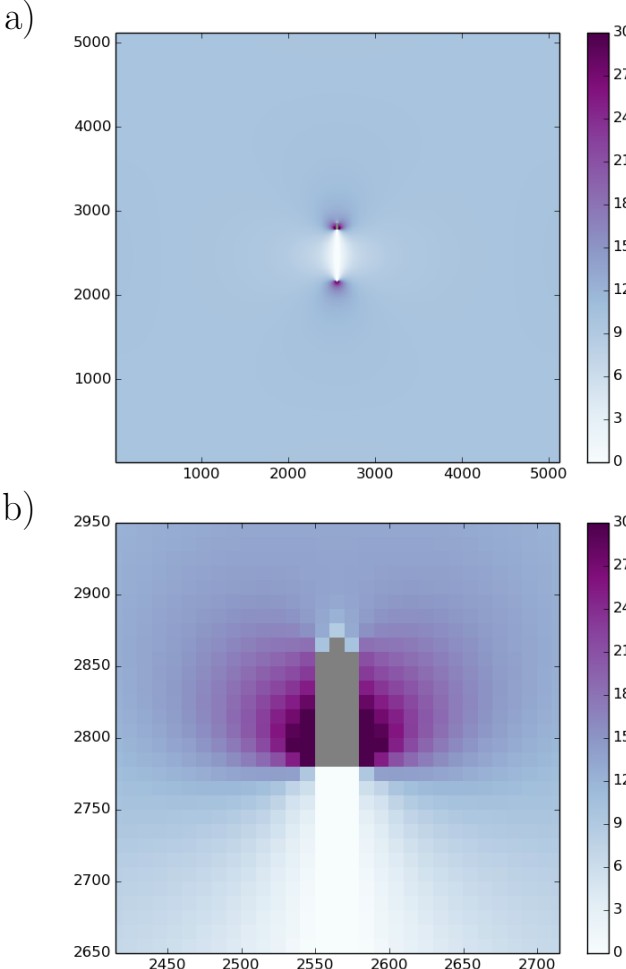

**Figure 9.** Pressure field at 10 m resolution when including a digitized ship 90 m long and 30 m wide (in gray). The thickness field is 2 m everywhere except a 600 m long lead behind the ship. The normal stresses at the boundaries are -10 kNm$^{-1}$. The whole domain is shown in panel a while panel b shows a zoom of the pressure field around the ship.


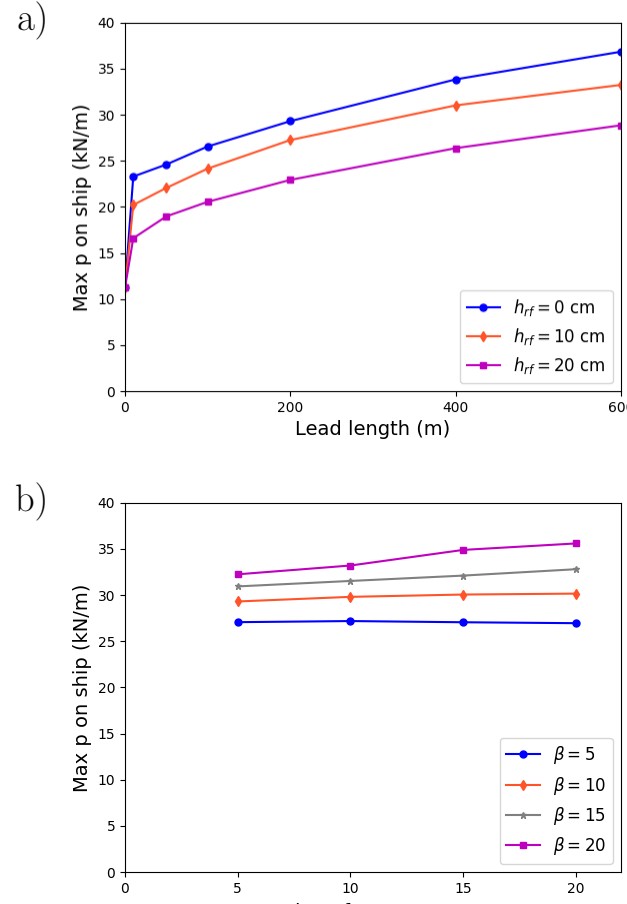

**Figure 10.** Maximum pressure (kNm$^{-1}$) on the ship as a function of the length of the lead behind the ship (a). The thickness field is 2 m everywhere except in the lead behind the ship; it is 0 cm for the blue curve, it is 10 cm for the orange one and it is 20 cm for the magenta one. Maximum pressure (kNm$^{-1}$) on the ship (b) as a function of the parameter $\kappa$ for four different values of the $\beta$ parameter ($\beta$=5: blue, $\beta$=10: orange, $\beta$=15: gray, $\beta$=20: magenta). The parameter $\kappa$ defines the shear strength on the contour of the ship while $\beta$ defines the strength in compression of the ship. The thickness field is 2 m everywhere except in the 200 m long lead behind the ship. The digitized ship is 90 m long and 30 m wide. The normal stresses at the boundaries are -10 kNm$^{-1}$.