# Peer review of "Toward a method for downscaling sea ice pressure"

_The Cryosphere, 2020_

## Referee Comment (RC1) · Anonymous Referee #1 · 19 May 2020

I think this paper is well-written, presents important new results on the downscaling of pack ice pressure in models for application to ships in ice, and should be accepted with only the following minor revision.

On Figures 4 and 5, panel (c): can the authors please specify what type of distribution was fitted to the data, provide the distribution parameters, and the 95% confidence intervals of the distribution parameters?

---

## Referee Comment (RC2) · Robert Frederking (Referee) · 10 Jun 2020

**Comments on paper tc-2020-134**

The work is correctly done, but it may be a little overly enthusiastic in applying the conclusions of the analysis to ship operations in ice. The problem analysed in the paper is one in engineering; the stress field around a void and/or inclusion in a large plate under stress. A ship moving through an ice field under pressure is a much more complex problem.

The work merits publication but the conclusion that the ship creates a stress concentration by breaking a channel might be couched in a more conditional manner. Experience generally shows that if the channel does not close, the ship is experiencing little or no pressure. The rate of channel closing and closing distance is proportional to the ice pressure that the ship feels. A longer open channel behind the ship is an indication of lower ice pressure, not higher.

Some specific corrections, improvements or comments:

- Larger font on some of the plots in figures would help readability.
- For Fig. 1 add the surface wind scale to panel b).
- Line 51; the author should be Loset?
- Line 175; would it complicate Fig. 2 to also show  $M_i$  and  $M_c$  on it?
- Line 211; Figure 4 a) and b) look very similar to results of finite element analysis of an elastic plate with a crack or void.
- Line 113; stress concentration at the tips of the lead and zero normal pressure on the boundary of the lead translate to the maximum and minimum pressures in Figure 4 c). I looks like the probability is greater than 1 for pressure 10 kN/m, check the y-axis scale. For the 10 m grid size the 28 cells that border on the lead versus the 5122 40 cells in the ice field give a 28/262104 (1.07E-04) probability of zero pressure. This doesn't seem to agree with Fig. 4 c).
- Figure 5 presents results of experiments with refrozen lead and ridged ice in addition to the 1 km lead. Not surprising is the result that there is no change of zero stress on the lead boundary or stress concentration at the tips of the lead. It seems that doubling the ice thickness from 1 to 2 m, Figure 4 versus Figure 5, the maximum stress at the tip of the leads is increased. Any explanation? Is it fair to compare maximum pressure in Fig. 6 b) with P\* = 20 kN/m2 for a 1 km long lead with Figure 5. Both are for 2 m ice thickness.
- For Figure 8 add a label to the x-axes, resolution and units of m. The maximum pressure of Figure 8 a) agrees with that in Figure 5 c), about 38 kN/m in both cases.
- The pressure field in Figure 9 seems reasonable given that a relatively stiff object (the ship) is placed at one end of a long cavity (lead). Your analysis only considers pressures, the ice also deforms and the further from the tip of the crack (lead) the greater the closing of the lead and thus higher lateral pressure.

The results presented in Figures 9 and 10 are quite consistent with the analysis model of a stiff object (ship) at the end of an elongated cavity (lead) in a more compliant medium (ice field). The results are consistent with stress analysis around inclusions. The analysis is correct, but it may be premature to draw conclusions about applying the results to operation of a ship in pressured ice.

There is literature in the Arctic engineering field that considers scale effect of ice pressures. The authors could look to this literature as they continue working in this field. See for example;

Sanderson, T.J.O., 1988. Ice Mechanics Risks to Offshore Structures. Graham and Trotman, London, UK.

Croasdale, K.R., 2009. Limit force ice loads – an update. Proceedings 20th POAC Conference, Paper POAC09-030, Lulea, Sweden.

---

## Referee Comment (RC3) · Harry Heorton (Referee) · 11 Jun 2020

**Toward a method for downscaling sea ice pressure**

The paper documents the development of an idealised sub-climate model grid cell (5kmx5km) modelling study of the sea ice internal pressures found at the tips of leads. A viscous plastic model is used to find the immediate internal stress states across the model for given internal stress states at the model domain boundaries. A single lead is placed within the sea ice of arbitrary size in the form of a rectangle of no ice and the stress states at the tips of the lead are documented. An idealised ship is placed at the end of the simulated lead and the simulated ice pressures on the ship are recorded. Model simulations are documented showing the changing ice stresses for a number of cases. First the model is tested for the case of no-ship, with the expected deformation rates related to an analytical case. Cases with leads of various sizes and for various model resolutions are also tested.

Additional ice features are also added to the domain, showing that the shape of the largest lead is the controlling factor for the highest ice pressures in the model. A ship is then positioned at the end of the largest lead and the stresses upon the ship are documented. Multiple experiments are performed varying the lead length (and also introducing a refrozen sea surface to the lead), ice strength parameter and the compressive and shear strengths of the ship itself. The authors conclude that the defects within a sub climate model grid cell are the greatest controller of sea ice pressure. They lead this conclusion to suggest that the pressure stress on a beset ship at the end of a lead of its own making will reduce as the lead surface consolidates. I can see how the results in this paper will help inform the navigation of ice covered seas.

The paper is in general very well written and the introduction and description are easy to follow. I suggest that is published with some additional explanations. Also the title of the paper show be changed to reflect the specific situation that is being simulated.

Improvements can be made to text in the form of overall motivation of the study. Explicitly saying in the introduction and methods and results that aim of the paper is to focus on the increased ice pressure at the tip of a lead where a ship is likely to be present would be a beneficial addition to the

paper. Also the paper needs to clearly state that this study models a single instantaneous stress field for a particular setup. This limitation also needs to be addressed in the conclusions when the case of lead closure is discussed. Whilst the authors mention that waiting for a lead to consolidate will reduce the stress on the ship, how likely is it that the lead will close mechanically before then? Also the authors state that care has been made to avoid all deformation within the model gird, what limitations does this put on the study? The authors mention that there is vast literature on ships navigating ice, does any of this describe the situation being simulated? In particular it would be helpful to discuss whether the modelled setup of a lead created by a ship within ice under uniform pressure, results in the lead remaining open and thus increased lead tip pressure existing as modelled here, is a likely and realistic scenario. I am not convinced that ice under uniform external pressure, when passed through by a ship will not result in lead closure, thus allowing the modelled setup to be encountered.

I find the results and numerical stability sections confusingly arranged. Further sub sectioning to break apart the various studies in the results will help. Collecting together all the cases where the model resolution was varied would be beneficial. After I had worked out what experiments had been performed and how they related to each I found them clear and well documented.

Title

I find that the title is not an accurate description of the paper content. The paper is focusing particularly on recreating the internal ice stresses at lead tips during constant ice compression for the case of ice stresses being low enough to not cause the closing of the lead. The paper content doesn't give a general method of downscaling as all the model setup is directly for the model case presented. The paper title should reflect this.

Abstract -

L6 Can you explain what form of numerical experiments you perform in this study within the abstract? A little extra depth on the nature of the methods used will be helpful here.

L10 The information within the parenthesis doesn't correspond well to the rest of the sentence. Do you mean that your study reveals that that the lead length is particularly important?

L13 I will be helpful here to clearly indicate that ice pressure is a horizontal 2d force.

L15 can you define 'ship besetting'

L16 predict the pressure field from what? using a force balance of applied wind, ocean stresses and sea ice drift.

L50 It might be helpful to include a brief introduction of previous square box ice modelling studies. VP simulations

https://www.researchgate.net/journal/1994-0440_The_Cryosphere_Discussions

More VP
Hutchings, J.K. et al. 2005. Modeling Linear Kinematic Features in Sea Ice. *Monthly Weather Review*. 133, 12 (Dec. 2005), 3481–3497. DOI:https://doi.org/10.1175/MWR3045.1.[1]

Using CICE
Heorton, H.D.B.S. et al. 2018. Stress and deformation characteristics of sea ice in a high-resolution, anisotropic sea ice model. *Phil. Trans. R. Soc. A*. 376, 2129 (Sep. 2018), 20170349. DOI:https://doi.org/10.1098/rsta.2017.0349.

[3]

Discrete element modelling
Wilchinsky, A.V. et al. 2010. Effect of shear rupture on aggregate scale formation in sea ice. *Journal of Geophysical Research*. 115, C10 (Oct. 2010). DOI:https://doi.org/10.1029/2009JC006043.

L114 equation 9. All previous equations are well described, Can you explain the physical reasoning for the replacement closure as well?

L149 An overview explanation here will make the following equations much easier to follow. From what I can tell you impose the total normal and shear stresses. The equations that follow enable you to give the components of the gradient of the internal stress tensor. Is this correct?

L159 does this mean that v(1m) will be solved for in the model? Can you list the components that need to be imposed for this side of the grid structure and those which will be left free?

L165 what happens to this simulation when the normal stress on the east and west side are not equal? I assume that there will be a large E-W ice drift which i understand is best avoided for your study.  This information will be very helpful for those who wish to recreate your model setup.

L168 This information doesn't require its own section, though including it is very useful. Perhaps put it with the coarse grain results, or in the previous or following section. Actually if all the methods are placed in a 'methods' section and subsections are used the paper format will be easier to navigate.

L175 so you have two masks - one defines the ship internal, one defines the ship contour.  On which contour iare the ice force balance equations solved? This section defines the technical boundary conditions of the mask, but a descriptive overview of what is done where on which mask will aid the readability of the technical description.

L181 does the ship ice strength imply that this ship is deformable? Was this done for realism or to allow the model to run effectively?

L183 Is the ships resistance to shear representing the shear strength at the ice/ship hull interface (so a form of friction between ice and steel) or the resistance the ship itself has to shearing? I guess the former as it seems as if the ship itself can not deform as it is fixed to the grid.

L189 ah you've mentioned the ice sliding around the ship. Do you apply different shear condition for each mask?

Section 7

Can you include some basic information about the model setup either here or back in section 3? What model simulations are you seeking? It seems that you are looking for static solutions, invariant in time, or a snap shot of ice stress, is this the case?  What are you hoping to show us with

these validations? You are comparing to idealised numbers of ice pressure. Do these validations show that the numerical model generates the correct pressures for a static field? For the lead cases presented here i was expecting to  see the closing of the lead, though this makes little sense if the simulations just show the immediate pressure field of ice with a lead present.

From reading ahead to the results it appear you are particularly interested in the increased stresses in the ice at the end of a lead, which a location where a ship is likely to be present. Informing the reader of this before the validation section will show why you are checking the pressure states to show that these regions are correctly simulated.

L207 how is it obtained from the model?

L222 what conclusions will you be seeking in the results section? The validations show that your model is good for the stress states you hope to test, but to fully show this you need to state what these stress states are and why the model and its setup work for them.

L350 my understanding is that the model gives the solution of a single 'snap-shot' of ice stress. The acceleration argument then surely does not matter?

---

## Author Comment (AC1) · 28 Jul 2020

**Response to reviewer 1**

We would like to thank reviewer 1 for his/her comments.

Note that based on comments from the reviewers, we have simplified and made the numerical experiments more uniform. First, the thickness of the level ice is 2 m for all the experiments. Second, the viscous coefficients (see eq. 5 and 6 in the revised manuscript) are always capped using the approach of Hibler 1979. Finally, the numerical approach was slightly modified: we seek the steady-state solution of $\rho h \partial u / \partial t = \nabla \cdot \sigma$. instead of solving directly $\nabla \cdot \sigma = 0$.

[Figure]

Although both approaches give the same answer, the new one is more consistent with the stability analysis described in the appendix. Because of these changes, all the numerical experiments were redone.

Below, the comments from the reviewers (1) are in normal character. Our responses (2) are in bold while changes to the manuscript (3) mentioned here are also in bold and in quotes.

REVIEWER 1

(1) I think this paper is well-written, presents important new results on the downscaling of pack ice pressure in models for application to ships in ice, and should be accepted with only the following minor revision.

On Figures 4 and 5, panel (c): can the authors please specify what type of distribution was fitted to the data, provide the distribution parameters, and the 95% confidence intervals of the distribution parameters?

**(2) Figures 4 and 5 are figures 3 and 4 in the revised manuscript. These figures show the probability density functions (PDF) calculated from the simulated 2D fields of pressure. There is no fit to the model outputs. The curves simply show the value of the PDF for all the bins (the bin size is 0.25 kNm$^{-1}$). We have added the following text when introducing figure 3:**

**(3) "From these simulated 2D pressure fields, probability density functions (PDF) are calculated using bins of 0.25 kNm$^{-1}$. They are shown in Fig. 3c which demonstrates that the simulated fields are very similar at 10 and 20 m resolutions."**
**Jean-François Lemieux**

---

## Author Comment (AC2) · 28 Jul 2020

**Response to reviewer 2**

We would like to thank reviewer 2 for his comments. Based on the comments from the reviewers, we have simplified and made the numerical experiments more uniform. First, the thickness of the level ice is 2 m for all the experiments. Second, the viscous coefficients (see eq. 5 and 6 in the revised manuscript) are always capped using the approach of Hibler 1979. Finally, the numerical approach was slightly modified: we seek the steady-state solution of  $\rho h \partial u / \partial t = \nabla \cdot \sigma$ , instead of solving directly  $\nabla \cdot \sigma = 0$ . Although both approaches give the same answer, the new one is more consistent with the stability analysis described in the

appendix. Because of these changes, all the numerical experiments were redone.

Reviewers 2 and 3 both had comments about mechanical closing of the lead behind the ship and that this should depend on the pressure at the boundaries (larger pressure should cause a shorter lead). To address these comments we have done additional experiments and added a new figure (Fig. 11). For the experiments of Fig.11, it is assumed that the length of the lead behind the ship decreases linearly as the pressure at the boundaries increases. Interestingly, we find that over a notable range of pressure applied at the boundaries, the maximum pressure on the ship does not vary much. This is a consequence of compensating effects: a larger pressure at the boundaries causes the lead to be shorter which decreases the stress concentration in the vicinity of the ship, making the maximum pressure weakly sensitive to the pressure at the boundary.

Below, the comments from the reviewers (1) are in normal character. Our responses (2) are in bold while changes to the manuscript (3) mentioned here are also in bold and in quotes.

**REVIEWER 2**

(1) The work is correctly done, but it may be a little overly enthusiastic in applying the conclusions of the analysis to ship operations in ice. The problem analysed in the paper is one in engineering; the stress field around a void and/or inclusion in a large plate under stress. A ship moving through an ice field under pressure is a much more complex problem.

(2) We agree. Note, however, that we do not consider the case of a ship moving through sea ice but only the case of a ship beset in heavy sea ice conditions. Note that we have added the following sentence in the introduction of the revised manuscript:
(3) "In contrast with studies mentioned in the last paragraph, we focus on ship besetting, rather than on a ship progressing in an ice covered region. We also study the downscaling of sea ice pressure from the km scale to scales relevant for navigation activities (tens of m)."

(1) The work merits publication but the conclusion that the ship creates a stress concentration by breaking a channel might be couched in a more conditional manner. Experience generally shows that if the channel does not close, the ship is experiencing little or no pressure. The rate of channel closing and closing distance is proportional to the ice pressure that the ship feels. A longer open channel behind the ship is an indication of lower ice pressure, not higher.

(2) We understand what the reviewer means here. But our point of view is that a ship beset might have a lead (i.e., a channel still open) behind it and that it is unclear what is the length of this lead. The numerical experiments with the ship should be seen as a sensitivity study about the impact of the lead length and ice conditions in the lead (which are unknowns). For the same large-scale pressure at the boundaries, we argue that the pressure on the ship should decrease as the lead closes (either thermodynamically or mechanically) behind the ship.

(2) To address this comment by the reviewer we have added an additional experiment for which it is assumed that the length of the lead decreases linearly as the large-scale pressure prescribed at the boundaries increases. This is described in subsection 6.2 and the results shown in a new figure (Fig. 11).

(1) Some specific corrections, improvements or comments:

(1) Larger font on some of the plots in figures would help readability.
**(2) We have reworked and improved all the figures.**

(1) For Fig. 1 add the surface wind scale to panel b).

(2) The reference vector is on the island on the lower-right side of the small domain.

(1) Line 51; the author should be Loset?

(2) Yes. It has been corrected.

(1) Line 175; would it complicate Fig. 2 to also show Mi and Mc on it?

(2) We have decided to simplify the way the digitized ship is defined. The ship is defined by land cells. The boundary conditions are no slip and no outflow. This is explained in the description of the experimental setup (section 3 in the revised manuscript). The masks Mi and Mc are not required anymore. Note that this leads to results qualitatively the same and allows us to draw the same conclusion.

(1) Line 211; Figure 4 a) and b) look very similar to results of finite element analysis of an elastic plate with a crack or void.

(1) Line 113; stress concentration at the tips of the lead and zero normal pressure on the boundary of the lead translate to the maximum and minimum pressures in Figure 4 c). I looks like the probability is greater than 1 for pressure 10 kN/m, check the y-axis scale. For the 10 m
grid size the 28 cells that border on the lead versus the 5122 - 40 cells in the ice field give a 28 / 262104 (1.07e-04) probability of zero pressure. This doesn't seem to agree with Fig. 4 c).

(2) This is due to the fact that we use small bins (of  $0.25 \text{ kNm}^{-1}$ ) and show the probability density not the probability. For figures 3 and 4, we have verified that the sum of the PDF times the bin width is indeed 1.0.

(1) Figure 5 presents results of experiments with refrozen lead and ridged ice in addition to the 1 km lead. Not surprising is the result that there is no change of zero stress on the lead boundary or stress concentration at the tips of the lead. It seems that doubling the ice thickness from 1 to 2 m, Figure 4 versus Figure 5, the maximum stress at the tip of the leads is increased. Any explanation? Is it fair to compare maximum pressure in Fig. 6 b) with  $P^* = 20 \text{ kN/m2}$  for a 1 km long lead with Figure 5. Both are for 2 m ice thickness.

(2) For both figures the thickness of the level ice is 2 m. The confusion is due to the fact that the validation experiment done just before the one for Fig. 4 was conducted with a constant thickness of 1 m. To improve the clarity of the manuscript, that experiment was redone with a thickness of 2 m. In fact, the thickness of the level ice is 2 m for all the experiments of the revised manuscript. Fig. 4 and Fig. 5 (Fig. 3 and Fig. 4 in the revised manuscript) do not show the same maximum pressure because the leads do not have the same width.

(1) For Figure 8 add a label to the x-axes, resolution and units of m. The maximum pressure of Figure 8 a) agrees with that in Figure 5 c), about 38 kN/m in both cases.

(2) Done.
(1) The pressure field in Figure 9 seems reasonable given that a relatively stiff object (the ship) is placed at one end of a long cavity (lead). Your analysis only considers pressures, the ice also deforms and the further from the tip of the crack (lead) the greater the closing of the lead and thus higher lateral pressure.

**(2) We agree. The limitations of our experimental setup are discussed in the conclusion of the revised manscript.**

(1) The results presented in Figures 9 and 10 are quite consistent with the analysis model of a stiff object (ship) at the end of an elongated cavity (lead) in a more compliant medium (ice field). The results are consistent with stress analysis around inclusions. The analysis is correct, but it may be premature to draw conclusions about applying the results to operation of a ship in pressured ice.

**(2) We have added a few sentences in the conclusion to describe the limitations of our numerical setup.**

(1) There is literature in the Arctic engineering field that considers scale effect of ice pressures. The authors could look to this literature as they continue working in this field. See for example;

Sanderson, T.J.O., 1988. Ice Mechanics Risks to Offshore Structures. Graham and Trotman, London, UK. Croasdale, K.

Croasdale, K.R., 2009. Limit force ice loads - an update. Proceedings 20th POAC Conference, Paper POAC09-030, Lulea, Sweden.
(2) We thank the reviewer for these references.

Jean-François Lemieux

---

## Author Comment (AC3) · 28 Jul 2020

**Response to reviewer 3**

We would like to thank reviewer 3 for his comments. Note that based on comments from the reviewers, we have simplified and made the numerical experiments more uniform. First, the thickness of the level ice is 2 m for all the experiments. Second, the viscous coefficients (see eq. 5 and 6 in the revised manuscript) are always capped using the approach of Hibler 1979. Finally, the numerical approach was slightly modified: we seek the steady-state solution of $\rho h \partial u / \partial t = \nabla \cdot \sigma$. instead of solving directly $\nabla \cdot \sigma = 0$. Although both approaches give the same answer, the new one is more consistent with the stability analysis described in the

appendix. Because of these changes, all the numerical experiments were redone.

Reviewers 2 and 3 both had comments about mechanical closing of the lead behind the ship and that this should depend on the pressure at the boundaries (larger pressure should cause a shorter lead). To address these comments we have done additional experiments and added a new figure (Fig. 11). For the experiments of Fig.11, it is assumed that the length of the lead behind the ship decreases linearly as the pressure at the boundaries increases. Interestingly, we find that over a notable range of pressure applied at the boundaries, the maximum pressure on the ship does not vary much. This is a consequence of compensating effects: a larger pressure at the boundaries causes the lead to be shorter which decreases the stress concentration in the vicinity of the ship, making the maximum pressure weakly sensitive to the pressure at the boundary.

Below, the comments from the reviewers (1) are in normal character. Our responses (2) are in bold while changes to the manuscript (3) mentioned here are also in bold and in quotes.

REVIEWER 3

(1) The paper documents the development of an idealised sub-climate model grid cell (5kmx5km) modelling study of the sea ice internal pressures found at the tips of leads. A viscous plastic model is used to find the immediate internal stress states across the model for given internal stress states at the model domain boundaries. A single lead is placed within the sea ice of arbitrary size in the form of a rectangle of no ice and the stress states at the tips of the lead are documented. An idealised ship is placed at the end of the simulated lead and the simulated ice pressures on the ship are recorded. Model simulations are documented showing the changing ice stresses for a number of cases. First the model is tested for the case of no-ship, with the expected deformation rates related to an analytical case. Cases with leads of various
sizes and for various model resolutions are also tested.

(1) Additional ice features are also added to the domain, showing that the shape of the largest lead is the controlling factor for the highest ice pressures in the model. A ship is then positioned at the end of the largest lead and the stresses upon the ship are documented. Multiple experiments are performed varying the lead length (and also introducing a refrozen sea surface to the lead), ice strength parameter and the compressive and shear strengths of the ship itself. The authors conclude that the defects within a sub climate model grid cell are the greatest controller of sea ice pressure. They lead this conclusion to suggest that the pressure stress on a beset ship at the end of a lead of its own making will reduce as the lead surface consolidates. I can see how the results in this paper will help inform the navigation of ice covered seas.

(1) The paper is in general very well written and the introduction and description are easy to follow. I suggest that is published with some additional explanations. Also the title of the paper should be changed to reflect the specific situation that is being simulated.

(1) Improvements can be made to text in the form of overall motivation of the study. Explicitly saying in the introduction and methods and results that aim of the paper is to focus on the increased ice pressure at the tip of a lead where a ship is likely to be present would be a beneficial addition to the paper. Also the paper needs to clearly state that this study models a single instantaneous stress field for a particular setup. This limitation also needs to be addressed in the conclusions when the case of lead closure is discussed.

**(2) The motivation of the study was improved/clarified in the revised manuscript. We have also changed the title of the paper (see below). To address the reviewer's comment about the instantaneous stress field, we have added the following sentence in section 3:**
(3) "Our numerical simulations therefore provide 2D static fields of the internal stresses inside this small domain."

(1) Whilst the authors mention that waiting for a lead to consolidate will reduce the stress on the ship, how likely is it that the lead will close mechanically before then?

(2) **Reviewer 2 had a similar comment. There is certainly mechanical closure but it is difficult to estimate the length of the lead for a certain large-scale pressure applied at the boundaries. This is why Fig. 10 should be viewed as the result of a sensitivity study. To address reviewers 2 and 3 comments about this we have conducted additional numerical experiments for which it was assumed that the length of the lead decreases when the pressure at the boundaries increases (see the new Fig.11).**

(1) Also the authors state that care has been made to avoid all deformation within the model grid, what limitations does this put on the study?

(2) **In the initially submitted manuscript the following text was included:**

(3) **"Also, in reality, sea ice convergence can cause ridging which can locally increase the yield strength of the ice. This strain hardening process was not considered in our numerical experiments; the maximum possible pressure in the domain is equal to $P^*h_l$."**

(2) **We have added a few additional sentences about the limitations of our numerical setup in the conclusion.**

(1) The authors mention that there is vast literature on ships navigating ice, does any of this

describe the situation being simulated?

**(2) In the revised manucript, we have improved the text presenting these other studies (see the introduction). We have also described better how our study is different than what was done by others. We have added the following sentences in the revised manuscript:**

**(3) "In contrast with studies mentioned in the last paragraph, we focus on ship besetting, rather than on a ship progressing in an ice covered region. We also study the downscaling of sea ice pressure from the km scale to scales relevant for navigation activities (tens of m)."**

**(3) "Idealized sea ice modeling studies with a continuum based approach have been conducted by specifying strain rates at the boundaries (e.g. Kubat et al. (2010); Ringeisen et al. (2019)) or by specifying wind patterns (e.g. Hutchings et al. (2005); Heorton et al. (2018)). However, to our knowledge, it is the first time that internal stresses are specified at the boundaries."**

(1) In particular it would be helpful to discuss whether the modelled setup of a lead created by a ship within ice under uniform pressure, results in the lead remaining open and thus increased lead tip pressure existing as modelled here, is a likely and realistic scenario. I am not convinced that ice under uniform external pressure, when passed through by a ship will not result in lead closure, thus allowing the modelled setup to be encountered.

**(2) We agree. Please see our other comment above about the new experiments and figure.**

(1) I find the results and numerical stability sections confusingly arranged. Further sub
sectioning to break apart the various studies in the results will help. Collecting together all the cases where the model resolution was varied would be beneficial. After I had worked out what experiments had been performed and how they related to each I found them clear and well documented.

**(2) We think the stability analysis should stay in an appendix as it is not essential for understanding this study. Following the reviewer's advice, we have added these two subsections in the result section:**

**(3) "Idealized sea ice experiments"**

**(3) "Experiments with an idealized ship"**

(1) Title I find that the title is not an accurate description of the paper content. The paper is focusing particularly on recreating the internal ice stresses at lead tips during constant ice compression for the case of ice stresses being low enough to not cause the closing of the lead. The paper content doesn't give a general method of downscaling as all the model setup is directly for the model case presented. The paper title should reflect this.

**(2) We think that the word "Toward" in the title indicates that we do not provide a complete method for downscaling the sea ice pressure. We have nevertheless changed the title so that it better reflects the fact that the goal of the method would be for navigation purposes. The new title is:**

**(3) "Toward a method for downscaling sea ice pressure for navigation purposes."**
(1) Abstract L6 Can you explain what form of numerical experiments you perform in this study within the abstract? A little extra depth on the nature of the methods used will be helpful here.

**(2) We have modified one sentence in the abstract in the revised manuscript. The sentence is:**

**(3) "In this paper, the downscaling of sea ice pressure from the km-scale to scales relevant for ships is investigated by conducting high resolution idealized numerical experiments with a viscous-plastic sea ice model"**

(1) L10 The information within the parenthesis does'nt correspond well to the rest of the sentence. Do you mean that your study reveals that that the lead length is particularly important?

**(2) The end of the abstract has been modified in the revised manuscript.**

(1) L13 I will be helpful here to clearly indicate that ice pressure is a horizontal 2d force.

**(2) We have added the following sentence in section 2:**

**(3) "As the stresses are vertically integrated, the stresses and stress invariants are 2D fields with units of $Nm^{-1}$"**

(1) L15 can you define "ship besetting"

**(2) We don't think this needs to be defined.**

(1) L16 predict the pressure field from what? using a force balance of applied wind, ocean stresses and sea ice drift.

**(2) The following sentence is in the revised manuscript:**

**(3) "By solving equations for the momentum balance and for the ice thickness distribution, sea ice models are able to predict the evolution of the pressure field.**

(1) L50 It might be helpful to include a brief introduction of previous square box ice modelling studies. VP simulations

More VP
Hutchings, J.K. et al. 2005. Modeling Linear Kinematic Features in Sea Ice. Monthly Weather Review. 133, 12 (Dec. 2005), 3481 - 3497.

Using CICE
Heorton, H.D.B.S. et al. 2018. Stress and deformation characteristics of sea ice in a high-resolution, anisotropic sea ice model. Phil. Trans. R. Soc. A. 376, 2129 (Sep. 2018), 20170349.

Discrete element modelling
Wilchinsky, A.V. et al. 2010. Effect of shear rupture on aggregate scale formation in sea ice. Journal of Geophysical Research. 115, C10 (Oct. 2010). DOI:https://doi.org/10.1029/2009JC006043.

**(2) We agree. We have added in the introduction the following text with some references:**

**(3) "Idealized sea ice modeling studies have been conducted by specifying strain rates at the boundaries (e.g. Kubat et al. (2010); Ringeisen et al.(2019)) or by specifying wind patterns (e.g. Hutchings et al. (2005); Heorton et al. (2018)). However, to our knowledge, it is the first time that internal stresses are specified at the boundaries."**

(1) L114 equation 9. All previous equations are well described, Can you explain the physical reasoning for the replacement closure as well?

**(2) We have added the following sentence:**

**(3) "The replacement pressure is commonly used in sea ice models to prevent unrealistic deformations of the sea ice cover when there is no external forcing."**

(1) L149 An overview explanation here will make the following equations much easier to follow. From what I can tell you impose the total normal and shear stresses. The equations that follow enable you to give the components of the gradient of the internal stress tensor. Is this correct?

**(2) We have added the following text in section 5:**

**(3) "In all the experiments, normal and shear stresses are applied at the four boundaries of the 5×5 km domain. For a given set of sea ice conditions, the steady-state solution of equation (10) is obtained. This provides us with the velocity field defined on the Arakawa C-grid. As the stresses and invariants are function of the sea ice conditions and velocity (see equations(2-9)), static 2D fields of the internal stresses and invariants are easily**

**obtained."**

(1) L159 does this mean that v(1m) will be solved for in the model? Can you list the components that need to be imposed for this side of the grid structure and those which will be left free?

**(2) We have added the following text in section 4:**

**(3) "Even though $u_{(1m)}$ is located at the boundary, it is solved along with $v_{(1m)}$ and all the other velocity components in the domain by the nonlinear solver."**

(1) L165 what happens to this simulation when the normal stress on the east and west side are not equal? I assume that there will be a large E-W ice drift which i understand is best avoided for your study. This information will be very helpful for those who wish to recreate your model setup.

**(2) The simulation blows up when the normal stress on the east and west sides are not equal. The reader is referred to the appendix for further explanations.**

(1) L168 This information doesn't require its own section, though including it is very useful. Perhaps put it with the coarse grain results, or in the previous or following section. Actually if all the methods are placed in a "methods" section and subsections are used the paper format will be easier to navigate.

**(2) We agree. As suggested, we have moved the figure to the result section.**

(1) L175 so you have two masks - one defines the ship internal, one defines the ship contour. On

which contour iare the ice force balance equations solved? This section defines the technical boundary conditions of the mask, but a descriptive overview of what is done where on which mask will aid the readability of the technical description.

**(2) We have decided to simplify the way the digitized ship is defined. The ship is defined by land cells. The boundary conditions are no slip and no outflow. This is explained in the description of the experimental setup (section 3 in the revised manuscript). The masks Mi and Mc are not required anymore. Note that this leads to results qualitatively the same and allows us to draw the same conclusion.**

(1) L181 does the ship ice strength imply that this ship is deformable? Was this done for realism or to allow the model to run effectively?

**(2) See our response above.**

(1) L183 Is the ships resistance to shear representing the shear strength at the ice/ship hull interface (so a form of friction between ice and steel) or the resistance the ship itself has to shearing? I guess the former as it seems as if the ship itself can not deform as it is fixed to the grid.

**(2) See our response above.**

(1) L189 ah you 've mentioned the ice sliding around the ship. Do you apply different shear condition for each mask?

**(2) See our response above.**
Section 7 (1) Can you include some basic information about the model setup either here or back in section 3? What model simulations are you seeking? It seems that you are looking for static solutions, invariant in time, or a snap shot of ice stress, is this the case? What are you hoping to show us with these validations? You are comparing to idealised numbers of ice pressure. Do these validations show that the numerical model generates the correct pressures for a static field? For the lead cases presented here i was expecting to see the closing of the lead, though this makes little sense if the simulations just show the immediate pressure field of ice with a lead present.

**(2) See our other responses above. We think we have improved the text in the introduction and in sections 3-6 of the revised manuscript. About the validation, the following sentence of the revised manuscript helps to understand how the pressure is simulated:**

**(3) "For a given set of sea ice conditions, the steady-state solution of equation (10) is obtained. This provides us with the velocity field defined on the Arakawa C-grid. As the stresses and invariants are function of the sea ice conditions and velocity (see equations (2-9)), static 2D fields of the internal stresses and invariants are easily obtained."**

(1) From reading ahead to the results it appear you are particularly interested in the increased stresses in the ice at the end of a lead, which a location where a ship is likely to be present. Informing the reader of this before the validation section will show why you are checking the pressure states to show that these regions are correctly simulated.

**(2) We have reworked the introduction. We think that the following sentences inform the reader about the ship experiments:**

**(3) "...we focus on ship besetting, rather than on a ship progressing in an ice covered region. We also study the downscaling of sea ice pressure from the km scale to scales relevant for navigation activities (tens of m)."**

**(3) "For our numerical experiments, we use a continuum based viscous-plastic sea ice model. In a first set of simulations, we study how the small-scale pressure depends on the stresses applied at the boundaries, on the ice conditions and on the rheology parameters. The second part of the results is dedicated to shipping applications; we investigate the small-scale pressure field in the vicinity of an idealized ship beset in heavy ice conditions and under compressive stresses."**

(1) L207 how is it obtained from the model?

**(2) See our other responses above. We think that the manuscript is clearer.**

(1) L222 what conclusions will you be seeking in the results section? The validations show that your model is good for the stress states you hope to test, but to fully show this you need to state what these stress states are and why the model and its setup work for them.

**(2) See our other responses above. We think we have addressed these points in the revised manuscript.**

(1) L350 my understanding is that the model gives the solution of a single "snap-shot" of ice stress. The acceleration argument then surely does not matter?

**(2) We understand that this was confusing. We have modified the text and have redone**

the simulations in order to find the steady-state solution of $\rho h \partial u / \partial t = \nabla \cdot \sigma$. **This is equivalent as solving for $\nabla \cdot \sigma = 0$ (both approaches give the same answer). This is now consistent with the stability analysis described in the appendix.**

**Jean-François Lemieux**

---

## Editor Decision (ED1)

**Toward a method for downscaling sea ice pressure**

The paper documents the development of an idealised sub-climate model grid cell (5kmx5km) modelling study of the sea ice internal pressures found at the tips of leads. A viscous plastic model is used to find the immediate internal stress states across the model for given internal stress states at the model domain boundaries. A single lead is placed within the sea ice of arbitrary size in the form of a rectangle of no ice and the stress states at the tips of the lead are documented. An idealised ship is placed at the end of the simulated lead and the simulated ice pressures on the ship are recorded. Model simulations are documented showing the changing ice stresses for a number of cases. First the model is tested for the case of no-ship, with the expected deformation rates related to an analytical case. Cases with leads of various sizes and for various model resolutions are also tested.

Additional ice features are also added to the domain, showing that the shape of the largest lead is the controlling factor for the highest ice pressures in the model. A ship is then positioned at the end of the largest lead and the stresses upon the ship are documented. Multiple experiments are performed varying the lead length (and also introducing a refrozen sea surface to the lead), ice strength parameter and the compressive and shear strengths of the ship itself. The authors conclude that the defects within a sub climate model grid cell are the greatest controller of sea ice pressure. They lead this conclusion to suggest that the pressure stress on a beset ship at the end of a lead of its own making will reduce as the lead surface consolidates. I can see how the results in this paper will help inform the navigation of ice covered seas.

The paper is in general very well written and the introduction and description are easy to follow. I suggest that is published with some additional explanations. Also the title of the paper show be changed to reflect the specific situation that is being simulated.

Improvements can be made to text in the form of overall motivation of the study. Explicitly saying in the introduction and methods and results that aim of the paper is to focus on the increased ice pressure at the tip of a lead where a ship is likely to be present would be a beneficial addition to the

paper. Also the paper needs to clearly state that this study models a single instantaneous stress field for a particular setup. This limitation also needs to be addressed in the conclusions when the case of lead closure is discussed. Whilst the authors mention that waiting for a lead to consolidate will reduce the stress on the ship, how likely is it that the lead will close mechanically before then? Also the authors state that care has been made to avoid all deformation within the model gird, what limitations does this put on the study? The authors mention that there is vast literature on ships navigating ice, does any of this describe the situation being simulated? In particular it would be helpful to discuss whether the modelled setup of a lead created by a ship within ice under uniform pressure, results in the lead remaining open and thus increased lead tip pressure existing as modelled here, is a likely and realistic scenario. I am not convinced that ice under uniform external pressure, when passed through by a ship will not result in lead closure, thus allowing the modelled setup to be encountered.

I find the results and numerical stability sections confusingly arranged. Further sub sectioning to break apart the various studies in the results will help. Collecting together all the cases where the model resolution was varied  would be beneficial. After I had worked out what experiments had been performed and how they related to each I found them clear and well documented.

Title

I find that the title is not an accurate description of the paper content. The paper is focusing particularly on recreating the internal ice stresses at lead tips during constant ice compression for the case of ice stresses being low enough to not cause the closing of the lead. The paper content doesn't give a general method of downscaling as all the model setup is directly for the model case presented. The paper title should reflect this.

Abstract -

L6 Can you explain what form of numerical experiments you perform in this study within the abstract? A little extra depth on the nature of the methods used will be helpful here.

L10 The information within the parenthesis doesn't correspond well to the rest of the sentence. Do you mean that your study reveals that that the lead length is particularly important?

L13 I will be helpful here to clearly indicate that ice pressure is a horizontal 2d force.

L15 can you define 'ship besetting'

L16 predict the pressure field from what? using a force balance of applied wind, ocean stresses and sea ice drift.

L50 It might be helpful to include a brief introduction of previous square box ice modelling studies. VP simulations

https://www.researchgate.net/journal/1994-0440_The_Cryosphere_Discussions

More VP
Hutchings, J.K. et al. 2005. Modeling Linear Kinematic Features in Sea Ice. *Monthly Weather Review*. 133, 12 (Dec. 2005), 3481–3497. DOI:https://doi.org/10.1175/MWR3045.1.[1]

Using CICE
Heorton, H.D.B.S. et al. 2018. Stress and deformation characteristics of sea ice in a high-resolution, anisotropic sea ice model. *Phil. Trans. R. Soc. A*. 376, 2129 (Sep. 2018), 20170349. DOI:https://doi.org/10.1098/rsta.2017.0349.

[3]

Discrete element modelling
Wilchinsky, A.V. et al. 2010. Effect of shear rupture on aggregate scale formation in sea ice. *Journal of Geophysical Research*. 115, C10 (Oct. 2010). DOI:https://doi.org/10.1029/2009JC006043.

L114 equation 9. All previous equations are well described, Can you explain the physical reasoning for the replacement closure as well?

L149 An overview explanation here will make the following equations much easier to follow. From what I can tell you impose the total normal and shear stresses. The equations that follow enable you to give the components of the gradient of the internal stress tensor. Is this correct?

L159 does this mean that v(1m) will be solved for in the model? Can you list the components that need to be imposed for this side of the grid structure and those which will be left free?

L165 what happens to this simulation when the normal stress on the east and west side are not equal? I assume that there will be a large E-W ice drift which i understand is best avoided for your study.  This information will be very helpful for those who wish to recreate your model setup.

L168 This information doesn't require its own section, though including it is very useful. Perhaps put it with the coarse grain results, or in the previous or following section. Actually if all the methods are placed in a 'methods' section and subsections are used the paper format will be easier to navigate.

L175 so you have two masks - one defines the ship internal, one defines the ship contour.  On which contour iare the ice force balance equations solved? This section defines the technical boundary conditions of the mask, but a descriptive overview of what is done where on which mask will aid the readability of the technical description.

L181 does the ship ice strength imply that this ship is deformable? Was this done for realism or to allow the model to run effectively?

L183 Is the ships resistance to shear representing the shear strength at the ice/ship hull interface (so a form of friction between ice and steel) or the resistance the ship itself has to shearing? I guess the former as it seems as if the ship itself can not deform as it is fixed to the grid.

L189 ah you've mentioned the ice sliding around the ship. Do you apply different shear condition for each mask?

Section 7

Can you include some basic information about the model setup either here or back in section 3? What model simulations are you seeking? It seems that you are looking for static solutions, invariant in time, or a snap shot of ice stress, is this the case?  What are you hoping to show us with

these validations? You are comparing to idealised numbers of ice pressure. Do these validations show that the numerical model generates the correct pressures for a static field? For the lead cases presented here i was expecting to  see the closing of the lead, though this makes little sense if the simulations just show the immediate pressure field of ice with a lead present.

From reading ahead to the results it appear you are particularly interested in the increased stresses in the ice at the end of a lead, which a location where a ship is likely to be present. Informing the reader of this before the validation section will show why you are checking the pressure states to show that these regions are correctly simulated.

L207 how is it obtained from the model?

L222 what conclusions will you be seeking in the results section? The validations show that your model is good for the stress states you hope to test, but to fully show this you need to state what these stress states are and why the model and its setup work for them.

L350 my understanding is that the model gives the solution of a single 'snap-shot' of ice stress. The acceleration argument then surely does not matter?

Comments on paper tc-2020-134

The work is correctly done, but it may be a little overly enthusiastic in applying the conclusions of the analysis to ship operations in ice. The problem analysed in the paper is one in engineering; the stress field around a void and/or inclusion in a large plate under stress. A ship moving through an ice field under pressure is a much more complex problem.

The work merits publication but the conclusion that the ship creates a stress concentration by breaking a channel might be couched in a more conditional manner. Experience generally shows that if the channel does not close, the ship is experiencing little or no pressure. The rate of channel closing and closing distance is proportional to the ice pressure that the ship feels. A longer open channel behind the ship is an indication of lower ice pressure, not higher.

Some specific corrections, improvements or comments:

- Larger font on some of the plots in figures would help readability.
- For Fig. 1 add the surface wind scale to panel b).
- Line 51; the author should be Loset?
- Line 175; would it complicate Fig. 2 to also show $M_i$ and $M_c$ on it?
- Line 211; Figure 4 a) and b) look very similar to results of finite element analysis of an elastic plate with a crack or void.
- Line 113; stress concentration at the tips of the lead and zero normal pressure on the boundary of the lead translate to the maximum and minimum pressures in Figure 4 c). I looks like the probability is greater than 1 for pressure 10 kN/m, check the y-axis scale. For the 10 m grid size the 28 cells that border on the lead versus the $512^2$ – 40 cells in the ice field give a 28/262104 (1.07E-04) probability of zero pressure. This doesn't seem to agree with Fig. 4 c).
- Figure 5 presents results of experiments with refrozen lead and ridged ice in addition to the 1 km lead. Not surprising is the result that there is no change of zero stress on the lead boundary or stress concentration at the tips of the lead. It seems that doubling the ice thickness from 1 to 2 m, Figure 4 versus Figure 5, the maximum stress at the tip of the leads is increased. Any explanation? Is it fair to compare maximum pressure in Fig. 6 b) with $P^* = 20$ kN/m$^2$ for a 1 km long lead with Figure 5. Both are for 2 m ice thickness.
- For Figure 8 add a label to the x-axes, resolution and units of m. The maximum pressure of Figure 8 a) agrees with that in Figure 5 c), about 38 kN/m in both cases.
- The pressure field in Figure 9 seems reasonable given that a relatively stiff object (the ship) is placed at one end of a long cavity (lead). Your analysis only considers pressures, the ice also deforms and the further from the tip of the crack (lead) the greater the closing of the lead and thus higher lateral pressure.

The results presented in Figures 9 and 10 are quite consistent with the analysis model of a stiff object (ship) at the end of an elongated cavity (lead) in a more compliant medium (ice field). The results are consistent with stress analysis around inclusions. The analysis is correct, but it may be premature to draw conclusions about applying the results to operation of a ship in pressured ice.

There is literature in the Arctic engineering field that considers scale effect of ice pressures. The authors could look to this literature as they continue working in this field.  See for example;

Sanderson, T.J.O., 1988. Ice Mechanics Risks to Offshore Structures. Graham and Trotman, London, UK.

[Figure]

Croasdale, K.R., 2009. Limit force ice loads – an update. Proceedings 20th POAC Conference, Paper POAC09-030, Lulea, Sweden.

[Figure]

I think this paper is well-written, presents important new results on the downscaling of pack ice pressure in models for application to ships in ice, and should be accepted with only the following minor revision.

On Figures 4 and 5, panel (c): can the authors please specify what type of distribution was fitted to the data, provide the distribution parameters, and the 95% confidence intervals of the distribution parameters?